# A Direct Link between Rényi–Tsallis Entropy and Hölder’s Inequality—Yet Another Proof of Rényi–Tsallis Entropy Maximization

**DOI:** 10.3390/e21060549

**Published:** 2019-05-30

**Authors:** Hisa-Aki Tanaka, Masaki Nakagawa, Yasutada Oohama

**Affiliations:** Graduate School of Informatics and Engineering, The University of Electro-Communications, Tokyo 182-8585, Japan

**Keywords:** Rényi–Tsallis entropy, generalized entropy, optimization, Hölder’s inequality

## Abstract

The well-known Hölder’s inequality has been recently utilized as an essential tool for solving several optimization problems. However, such an essential role of Hölder’s inequality does not seem to have been reported in the context of generalized entropy, including Rényi–Tsallis entropy. Here, we identify a direct link between Rényi–Tsallis entropy and Hölder’s inequality. Specifically, we demonstrate yet another elegant proof of the Rényi–Tsallis entropy maximization problem. Especially for the Tsallis entropy maximization problem, only with the equality condition of Hölder’s inequality is the *q*-Gaussian distribution uniquely specified and also proved to be optimal.

## 1. Introduction

Tsallis entropy [1,2] has been recently utilized as a versatile framework for expanding the realm of Shannon–Boltzmann entropy for nonlinear processes, in particular, those that exhibit power–law behavior. It shares a structure in common with Rényi entropy [3], Daróczy entropy [4], and probability moment presented in Moriguti [5], since the essential part of all these functionals is $\int p^{q}\left(x\right)dx$ (or $\sum p_{i}^{q}$) for certain constrained probability density functions p(x) (or $p_{i}$). This naturally has been of interest for a variety of issues in information theory and related areas. For instance, in his pioneering work, Campbell [6] stated that “Implicit in the use of average code length as a criterion of performance is the assumption that cost varies linearly with code length. This is not always the case.” Then, Campbell [6] introduced a nonlinear average length measure defined as
$$\begin{array}{c} L\left(t\right)=\frac{1}{t}\log_{D}\sum_{i=1}^{N}p_{i}D^{tl_{i}}, \end{array}$$
being an extension of the one by Shannon,
$$\begin{array}{c} L_{0}=\sum_{i=1}^{N}p_{i}l_{i}, \end{array}$$
in which *D* is the size of the alphabet, $p_{i}$ is the probability for a source to produce symbols $x_{i}$, $l_{i}$ is the length of a codeword $c_{i}$ mapped from symbol $x_{i}$ (using *D* letters of the alphabet) in the context of source coding, and *t* is an arbitrary parameter (0<t<∞). One of the surprising facts proved in [6] is that the lower bound to the moment-generating function of code lengths, namely, L(t), is given by $H_{\frac{1}{1+t}}\left(p\right)$, namely, Rényi entropy of order ${\left(1+t\right)}^{-1}$ of the source $p={\left\{p_{i}\right\}}_{i=1}^{N}$. Moreover, Ref. [6] also realizes that, if
$$\begin{array}{c} l_{i}=\frac{1}{1+t}\log_{D}\frac{1}{p_{i}}+\frac{t}{1+t}H_{\frac{1}{1+t}}\left(p\right), \end{array}$$
which is a mixture of the Shannon code length $\log_{D}\frac{1}{p_{i}}$ and Rényi entropy of order ${\left(1+t\right)}^{-1}$, we have the lower bound $L\left(t\right)=H_{\frac{1}{1+t}}\left(p\right)$. So far, Baer [7] has further generalized this result and constructed an algorithm for finding optimal binary codes under quasiarithmetic penalties. In addition, new extensions of [6] were obtained by Bercher [8] and by Bunte and Lapidoth [9].

Such an instance, where “a nonlinear measure” (i.e., generalized entropy) naturally arises, is also known for channel capacities. Daróczy [4] first analyzed a generalized channel capacity, which is a natural consequence of his extension of Shannon entropy (i.e., Daróczy entropy). This result has initiated extensive work in this direction. For instance, Landsberg and Vedral [10] first introduced Rényi entropy and Tsallis entropy for a binary symmetric channel, and they suggested the possibility of “super-Shannon channel capacities.” More recently, Ilić, Djordjević, and Küeppers [11] obtained new expressions for generalized channel capacities by introducing Daróczy–Tsallis entropy even for a weakly symmetric channel, binary erasure channel, and *z*-channel. Similar extensions have been explored for rate distortion theory. For instance, Venkatesan and Plastino [12] developed nonextensive rate distortion theory by introducing Tsallis entropy and constructed a minimization algorithm for generalized mutual information. More recently, Girardin and Lhote [13] covered the setting in [12] in a general framework of generalized entropy rates, which includes Rènyi–Tsallis entropy.

In the context of generalized entropy just described, the *q*-Gaussian distribution [1,2] often emerges as a maximizer of Rényi–Tsallis entropy under certain constraints, and, hence, it has been extensively studied. Since the *q*-Gaussian effectively models power–law behavior with a one-parameter *q*, its utility is widespread in various areas, including new random number generators proposed by Thistleton, Marsh, Nelson, and Tsallis [14] and by Umeno and Sato [15]. In addition to such an important application in communication systems, queuing theory has recently incorporated the *q*-Gaussian, reflecting the heavy-tailed traffic characteristics observed in broadband networks [16,17,18,19]. For instance, Karmeshu and Sharma [16] introduced Tsallis entropy maximization, and, there, the *q*-Gaussian emerges as the queue length distributions, which suggests that Jaynes’ maximum entropy principle [20,21,22] can be generalized to a framework of Tsallis entropy.

Some of the above issues are formulated as nonlinear optimizations with “a nonlinear measure” under certain constraints (which depend on each issue). As mentioned above, Rényi–Tsallis entropy and *q*-Gaussian is one such instance. In other words, the *q*-Gaussian maximizes Tsallis entropy under certain constraints. Therefore, it is useful to obtain a deeper understanding of such nonlinear optimization problems. In this study, we find a direct link between Rényi–Tsallis entropy and Hölder’s inequality that leads to yet another elegant proof of Rényi–Tsallis entropy maximization. The idea of the proof is different from those offered in previous studies (for instance, [23,24,25,26,27]) as explained below. Interestingly, the technique developed in this study might possibly be useful for tackling more complicated problems regarding optimization issues in information theory and other research areas, such as the conditional Rényi entropy (as in [28,29,30]), for instance.

Previous studies [23,24,25,26,27] are based on a common standpoint, the generalization of the moment–entropy inequality (cf. [25,26]). Namely, they intend to generalize the situation that a continuous random variable with a given second moment and maximal Shannon entropy is a Gaussian distribution (cf. [3], Theorem 8.6.5). In doing so, a *generalized relative entropy* is devised, which takes a different form (and has a different name) depending on the problem. First of all, Tsukada and Suyari’s beautiful work [23] has given proofs for Rényi entropy maximization, which is also known as a bound of Moriguti’s probability moment [5] (as posed in **R1** in Section 2). Namely, they prove that the *q*-Gaussian distribution [1,2] is a unique optimal solution by utilizing the fact that all feasible solutions constitute a convex set. Although [23] does not explicitly construct a generalized relative entropy, the essential structure of the proofs inherits the one in the proof of the moment-entropy inequality ([3], Theorem 8.6.5)).

Moreover, they have identified an explicit one-to-one correspondence between feasible solutions to the problems of Rényi entropy maximization and Tsallis entropy maximization, which is also shown in ([31], p. 754). This implies that an ‘indirect’ proof to Tsallis entropy maximization (as posed in **T1** in Section 2) has been first obtained in [23]. In contrast to this proof, the first ‘direct’ proof to Tsallis entropy maximization is obtained in Furuichi’s elegant work [24]. The proof in [24] utilizes nonnegativity of the *Tsallis relative entropy* defined between the *q*-Gaussian distribution (i.e., a possible maximizer) and any other feasible solution. On the other hand, the remarkable work of Lutwak, Yang, and Zhang first clarified that *generalized Gaussians* maximize λ-*Rényi entropy power* under a constraint on the *p*-th moment of the distribution, for univariate distributions [25] and for the associated *n*-dimensional extensions [26].The essential point in the proofs in [25,26] is construction of *relative*
λ-*Rényi entropy power*, which is nonnegative and takes a quite different form compared to the *Tsallis relative entropy* in [24]. (More precisely, in [25], they prove nonnegativity of the relative λ-*Rényi entropy*
$\log N_{\lambda }\left[f,g\right]$ ([25], Lemma 1). Starting from this nonnegativity:$\log N_{\lambda }\left[f,Gt\right]\geq 1$, they construct a series of inequalities that saturate at the generalized Gaussian ([25], Lemma 2). Note, however, that, as observed in this $N_{\lambda }\left[f,Gt\right]$, they start by giving a candidate of the maximizer ab initio, which is the generalized Gaussian Gt.) Furthermore, Vignat, Hero, and Costa [32] obtained a general, sharp result using the *Bregman information divergence* for an *n*-dimensional extension of Tsallis entropy. In addition to [25,26,32], Eguchi, Komori, and Kato’s interesting results [27] include the same *n*-dimensional extension to Tsallis entropy. (Ref. [32] has also identified an elegant structure regarding the projective divergence and the γ-loss functions in maximum likelihood estimation.)Similar to [24,25,26,32], the key component of the proof in [27] is the *projective power divergence,* which again takes a quite different form compared to the ones in [24,25,26,32]. To prove nonnegativity of the generalized relative entropy, Refs. [25,26,27] utilize Hölder’s inequality, but Refs. [23,24,32] do not. Namely, Hölder’s inequality has been an auxiliary useful tool, and it has never played an essential role in these previous studies. In addition to the construction of generalized relative entropies, the optimal *q*-Gaussian distribution needs to be ‘given ab initio’ [23,24,25,26,27,32], inheriting the framework showing that the Gaussian distribution maximizes Shannon entropy ([3], Theorem 8.6.5).

Now natural questions arise: is it possible to systematically solve the problems of Rényi–Tsallis entropy maximization in a different (and hopefully simpler) way than the previous study? In addition, is it possible to ‘construct’ the *q*-Gaussian distribution? These questions are positively answered from a new viewpoint as follows. First, only by the equality (i.e., saturation) condition of Hölder’s inequality, the *q*-Gaussian distribution is specified, and, at the same time, its optimality is proved by Hölder’s inequality for a Tsallis entropy maximization of 1<q<3 (Theorem 1) and of 0≤q<1 (Theorems 2 and 3). This clarifies how and why the *q*-Gaussian distribution emerges as the maximizer in an explicit way for the first time in the literature. (To the authors’ knowledge, such a characterization of the *q*-Gaussian distribution has never been reported.) However, for a Rényi entropy maximization of q>1 (Theorem 4) and of $\frac{1}{3}<q<1$ (Theorem 5), the *q*-Gaussian distribution is specified with the aid of the equality condition of Hölder’s inequality. In addition, the proof of its optimality requires a simple inequality inspired from Moriguti [5]. Note that we do not intend to provide an explicit characterization of the *q*-Gaussian distribution in terms of the parameter *q*, since numerous previous studies (including [23,24,25,26,27]) have already clarified this. Nevertheless, regarding Tsallis entropy maximization when q=0, which has previously been studied in [2], a rigorous result (as in Theorem 3) is now obtained for the first time thanks to Hölder’s inequality. (For instance, in the framework of [24], the case for q=0 cannot be incorporated because the Tsallis relative entropy is not defined adequately.)

We note that Hölder’s inequality has been recently utilized as an essential tool for optimization in Campbell [6], Bercher [8], and Bunte and Lapidoth [9]; on source coding, in Bercher [33,34]; on generalized Cramér–Rao inequalities; and in Tanaka [35,36] on a physical limit of injection locking. However, such an essential role of Hölder’s inequality does not seem to be reported in the context of generalized entropy, including Rényi entropy (cf. [37]), except for the use as a means for proving nonnegativity of a generalized relative entropy, as mentioned above.

In what follows, Section 2 introduces basic definitions required for the analysis. Section 3 includes the main results regarding Rényi–Tsallis entropy maximization problems, and it also contains an explanation on the link to Moriguti’s argument in [5]. Section 4 lists the proofs to the results presented in Section 3. Finally, one Appendix A at the end provides further supplementary information.

## 2. Basic Definitions and Problem Formulation

In this section, we first define Tsallis entropy [1,24] and Rényi entropy ([3], pp. 676–679). Next, we reformulate Rényi–Tsallis entropy maximization problems in a unified way. Finally, we introduce Hölder’s inequality in relation to the problems in this study.

### 2.1. Tsallis Entropy and Rényi Entropy

Tsallis entropy is beautifully presented in the context of *q*-analysis (cf. [1], p. 41) as follows. First, the *q*-exponential function $\exp_{q}$, whose domain and range satisfies
$$\begin{array}{c} \exp_{q}:\left\{\begin{array}{cc} \left[\frac{-1}{1-q},\infty \right)\to R^{+}\cup \left\{0\right\} & \mathrm{if}\;q<1,\\ \left(-\infty ,\frac{1}{q-1}\right]\to R^{+} & \mathrm{if}\;q>1, \end{array}\right. \end{array}$$
is defined by
$$\begin{array}{c} \exp_{q}x={\left[1+\left(1-q\right)x\right]}^{\frac{1}{1-q}}. \end{array}$$

While, the inverse of the *q*-exponential function, namely *q*-logarithmic function, is defined by
$$\begin{array}{c} \ln_{q}x=\frac{x^{1-q}-1}{1-q}. \end{array}$$

Note that, as q→1, we have $\exp_{q}x\to e^{x}$ and $\ln_{q}x\to \ln x$. We also note that the above definition of $\exp_{q}x$ and $\ln_{q}x$ has been recently revised by Oikonomou and Bagci [38]. (In [38], they have further developed ‘complete’ *q*-exponentials and *q*-logarithms.) Then, the Tsallis entropy $H_{q}^{\mathrm{Tsallis}}$ is defined by
(1)$$\begin{array}{c} H_{q}^{\mathrm{Tsallis}}\left[p\right]\;=\;-\int_{-\infty }^{\infty }p^{q}\left(x\right)\ln_{q}p(x)\;dx\;=-⟪p^{q}\left(x\right)\ln_{q}p(x)⟫, \end{array}$$
for univariate probability density functions (PDFs) *p* on R, which is a natural generalization of Boltzmann–Gibbs entropy and Shannon entropy. Hereafter, $⟪\cdot ⟫=\int_{-\infty }^{\infty }\cdot \;dx$ in (Equation 1) is used for notational simplicity. The reason why $⟪\cdot ⟫$ is used, instead of $\langle \cdot \rangle$, is due to the fact that $\langle \cdot \rangle$ is generally used for the expectation value. On the other hand, Rényi entropy is well-known and can be found in textbooks of information theory (cf. [3], pp. 676–679), which is defined simply by
$$H_{q}^{Ré\mathrm{nyi}}\left[p\right]=\frac{\ln⟪p^{q}\left(x\right)⟫}{1-q}\;\;\left(0<q<\infty \;\;\left(q\neq 1\right)\right).$$

Finally, we note that only differential entropies (i.e., continuous probability distributions) are considered in this study, although our technique with Hölder’s inequality can be applied to discrete probability distributions.

### 2.2. Problem Formulation

Let D be the set of all PDFs on R. We then define the set, as introduced in [24],
$$\begin{array}{c} C_{q}=\left\{p\;|\;p\in D\;\mathrm{and}\;\frac{⟪x^{2}p^{q}\left(x\right)⟫}{⟪p^{q}\left(x\right)⟫}<\infty \right\}\;\left(\subset D\right). \end{array}$$

Following the problem formulation in [1,2,23,24,31], we first introduce the Tsallis entropy maximization problem for univariate PDFs *p* on R:
(2a)$$\begin{array}{cc} T1: & \;\underset{p\;\in \;C_{q}}{\mathrm{maximize}\;\;}H_{q}^{\mathrm{Tsallis}}\left[p\right]\;=\;-⟪p^{q}\left(x\right)\;\ln_{q}p\left(x\right)⟫=\frac{1-⟪p^{q}\left(x\right)⟫}{q-1}\;\;\left(\mathrm{for}\;\;q\;\geq \;0\;\;(q\neq 1)\right)\; \end{array}$$
(2b)$$\begin{array}{c} \;\mathrm{subject}\;\mathrm{to}\;\;⟪p(x)⟫=1,\; \end{array}$$
(2c)$$\begin{array}{c} \;\frac{⟪x^{2}p^{q}\left(x\right)⟫}{⟪p^{q}\left(x\right)⟫}\;=⟪x^{2}\tilde{P_{q}}\left(x\right)⟫=\sigma^{2},\; \end{array}$$
in which *q* and $\sigma^{2}$ have fixed values, and $\tilde{P_{q}}\left(x\right)\;=\;\;p^{q}\left(x\right)/⟪p^{q}\left(x\right)⟫$. Note that $⟪p^{q}\left(x\right)⟫<\infty$ and $⟪x^{2}p^{q}\left(x\right)⟫<\infty$ are assumed in **T1**. ${\tilde{P}}_{q}\left(x\right)$ is often called the escort probability [27,31]. This somewhat unusual form of expectation $⟪x^{2}{\tilde{P}}_{q}\left(x\right)⟫=\sigma^{2}$ is called the *q*-normalized expectation [31], which has been usually assumed in Tsallis statistics. In contrast to the *q*-normalized expectation, as [31] pointed out, the usual expectation $⟪x^{2}p\left(x\right)⟫=\sigma^{2}$ is also valid in Tsallis statistics. We note the Tsallis entropy maximization problem under the constraint of this usual expectation is considered later in problem **R2**.

For problem **T1**, using Tsallis relative entropy, Furuichi [24] first proved that for 0<q<3 the *q*-Gaussian distribution $p\left(x\right)=\frac{1}{Z_{q}}\exp_{q}\left(-\beta_{q}x^{2}\right)$ maximizes the Tsallis entropy among any univariate PDFs in $C_{q}$, where $Z_{q}\;$ and $\;\beta_{q}$ are constants determined by *q* and σ.

Here, we formulate a slightly generalized optimization problem **T2**, as follows. First, replace (2c) with
$$\begin{array}{c} ⟪x^{2}p^{q}\left(x\right)⟫-\sigma^{2}⟪p^{q}\left(x\right)⟫=\;⟪\left(x^{2}-\sigma^{2}\right)p^{q}\left(x\right)⟫=0. \end{array}$$

Note that now, as opposed to **T1**, it is not necessarily required that both $⟪x^{2}p^{q}\left(x\right)⟫$ and $⟪p^{q}\left(x\right)⟫$ are finite, and hence, $C_{q}$ is not required, and it is replaced with D. Next, notice that Tsallis entropy is maximal at p(x), such that $⟪p^{q}\left(x\right)⟫$ is minimal (or correspondingly, maximal at p(x), such that $⟪p^{q}\left(x\right)⟫$ is maximal) for q>1 (correspondingly, for 0≤q<1). Then, by introducing an additional arbitrary parameter $\lambda_{q}$, **T1** is reformulated as
$$\begin{array}{cc} \;T2: & \;\underset{p\;\in \;D}{\mathrm{minimize}}\;\left(\mathrm{or}\;\mathrm{maximize}\right)\;\;T_{q}\left[p;\lambda_{q}\right]\;=\;\sigma^{2}⟪p^{q}\left(x\right)⟫+\lambda_{q}⟪\left(x^{2}-\sigma^{2}\right)p^{q}\left(x\right)⟫\; \end{array}$$
(3a)$$\begin{array}{c} \;=⟪p^{q}\left(x\right)\left[\lambda_{q}x^{2}+\left(1-\lambda_{q}\right)\sigma^{2}\right]⟫\;\;\;\left(\lambda_{q}\in R,\;\mathrm{for}\;q>1\;\left(\mathrm{correspondingly},\;\;\mathrm{for}\;\;0\leq q<1\right)\right)\; \end{array}$$
(3b)$$\begin{array}{c} \;\mathrm{subject}\;\mathrm{to}\;\;⟪p(x)⟫=1,\; \end{array}$$
(3c)$$\begin{array}{c} \;⟪\left(x^{2}-\sigma^{2}\right)p^{q}\left(x\right)⟫=0,\; \end{array}$$
where the constant $\sigma^{2}$ is multiplied with $⟪p^{q}\left(x\right)⟫$ in the first term of ([Disp-formula FD3a-entropy-21-00549]) simply due to notational convenience for later analysis in Section 4.

As opposed to the Tsallis entropy maximization problem **T1**, the Rényi entropy maximization problem is usually considered under the constraint of the usual expectation $⟪x^{2}p\left(x\right)⟫=\sigma^{2}$, in other words,
(4a)$$\begin{array}{cc} \;R1: & \;\mathrm{maximize}\;\;H_{q}^{Ré\mathrm{nyi}}\left[p\right]=\frac{\ln⟪p^{q}\left(x\right)⟫}{1-q}\;\;\left(\mathrm{for}\;0<q<\infty \;\left(q\neq 1\right)\right)\; \end{array}$$
(4b)$$\begin{array}{c} \;\mathrm{subject}\mathrm{to}\;\;⟪p(x)⟫=1,\; \end{array}$$
(4c)$$\begin{array}{c} \;⟪x^{2}p\left(x\right)⟫=\sigma^{2},\; \end{array}$$
which is equivalent to:
(5a)$$\begin{array}{c} \;\underset{p\;\in \;D}{\mathrm{minimize}}\;(\mathrm{or}\;\mathrm{maximize})\;\;⟪p^{q}\left(x\right)⟫\;\;(q>1\;(\mathrm{correspondingly},\;0\;\leq \;q<1))\;\; \end{array}$$
(5b)$$\begin{array}{c} \;\mathrm{subject}\;\mathrm{to}\;\;⟪p(x)⟫=1,\; \end{array}$$
(5c)$$\begin{array}{c} \;⟪x^{2}p\left(x\right)⟫=\sigma^{2}.\; \end{array}$$

We note this very problem for q>1 was first posed and solved by Moriguti in 1952 [5]. (Later in [39], cases q>1 and 0<q<1 are both analyzed in an *n*-dimensional spherical symmetric extension of [5] with the same approach as [5].)Similar to **T1**, by introducing an additional parameter $\lambda_{q}$ and the constraint
(6)$$\begin{array}{c} ⟪x^{2}p\left(x\right)⟫-\sigma^{2}⟪p(x)⟫=⟪\left(x^{2}-\sigma^{2}\right)p\left(x\right)⟫=0, \end{array}$$
which is obtained from (5b) and (5c), **R1** is now reformulated as
$$\begin{array}{cc} R2: & \;\underset{p\;\in \;D}{\mathrm{minimize}}\;\left(\mathrm{or}\;\mathrm{maximize}\right)\;\;R_{q}\left[p;\lambda_{q}\right]\;=\;⟪p^{q}\left(x\right)⟫+\lambda_{q}⟪\left(x^{2}-\sigma^{2}\right)p\left(x\right)⟫\; \end{array}$$
(7a)$$\begin{array}{cc} & \;=⟪p^{q}\left(x\right)\left[1+\lambda_{q}\left(x^{2}-\sigma^{2}\right)p^{1-q}\left(x\right)\right]⟫\;\;\;\left(\mathrm{for}\;q>1\;(\mathrm{correspondingly},\;\mathrm{for}\;\;0\;\leq \;q<1),\lambda_{q}\in R\right)\; \end{array}$$
(7b)$$\begin{array}{c} \;\mathrm{subject}\;\mathrm{to}\;\;⟪p(x)⟫=1,\; \end{array}$$
(7c)$$\begin{array}{c} \;⟪\left(x^{2}-\sigma^{2}\right)p\left(x\right)⟫=0.\; \end{array}$$

As we observe ([Disp-formula FD3a-entropy-21-00549]) in **T2** and ([Disp-formula FD7a-entropy-21-00549]) in **R2**, both become the inner products of two functions; $p^{q}\left(x\right)$ and $\lambda_{q}x^{2}+\left(1-\lambda_{q}\right)\sigma^{2}$ and $p^{q}\left(x\right)$ and $1+\lambda_{q}\left(x^{2}-\sigma^{2}\right)p^{1-q}\left(x\right)$, respectively. This suggests a direct link to Hölder’s inequality.

### 2.3. Hölder’s Inequality for Later Analysis

Here, we provide minimum information about Hölder’s inequality for later analysis in Section 3 and Section 4. The standard Hölder’s inequality is given by
(8)$$\begin{array}{c} {∥fg∥}_{1}\leq {∥f∥}_{\alpha }{∥g∥}_{\beta }, \end{array}$$
with 1≤α, β≤∞ and $\alpha^{-1}+\beta^{-1}=1$ (cf. [40] for the one-demensional case and [41] for general measurable functions). In general, *f* and *g* are measurable functions defined on a subset $S\subseteq R^{n}$ and μ(S)>0, and we employ a compact notation as
$$\begin{array}{ccc} {∥f∥}_{\alpha } & = & {\left(\int_{S}{\left|f\right|}^{\alpha }\;d\mu \right)}^{\frac{1}{\alpha }},\\ {∥g∥}_{\beta } & = & {\left(\int_{S}{\left|g\right|}^{\beta }\;d\mu \right)}^{\frac{1}{\beta }}. \end{array}$$

Although ${∥\cdot ∥}_{\alpha }$ and ${∥\cdot ∥}_{\beta }$ are no longer norms for α,β<1, now in the context of this study, we set $\alpha =q^{-1}$ and $\beta ={\left(1-q\right)}^{-1}$. Then, Hölder’s inequality (Equation 8) is given in the following form:(9)$$\begin{array}{c} {∥fg∥}_{1}\;\leq \;{∥f∥}_{\frac{1}{q}}{∥g∥}_{\frac{1}{1-q}}\;\;\left(0\leq q\leq 1\right). \end{array}$$

For the case 0<q<1, the equality in (Equation 9) holds if and only if there exists constants *A* and *B*, not both 0 (cf. [40], p. 140), (More specifically, if *f* is null (i.e., f(s)=0(a.e.s∈S)), then B=0. In addition, if *g* is null, A=0.) such that
(10)$$\begin{array}{c} A{\left|f(s)\right|}^{\frac{1}{q}}=B{\left|g(s)\right|}^{\frac{1}{1-q}}\;\;(a.e.\;s\in S). \end{array}$$
In addition, for the exceptional case q=0 (as well as q=1), we can argue a condition for the equality in (Equation 9) separately, as shown in Section 4.3, although the expression of (Equation 10) is no more valid for this case.

In contrast to (Equation 9), reverse Hölder’s inequality is given by
(11)$$\begin{array}{c} {∥fg∥}_{1}\;\geq \;{∥f∥}_{\frac{1}{q}}{∥g∥}_{\frac{-1}{q-1}}\;\;\left(q>1\right), \end{array}$$
which is directly obtained from Hölder’s inequality [40]. We note that *f* can be 0 over any subset U⊆S. As for *g*, on the other hand, we assume g(s)≠0 for almost everywhere (a.e.) s∈S, taking care that $\frac{-1}{q-1}<0$ in (Equation 11) (cf. [40], p. 140). Then, for the case q>1, the equality in (Equation 11) holds if and only if there exists A≥0, such that
(12)$$\begin{array}{c} \left|f(s)\right|=A{\left|g(s)\right|}^{\frac{-q}{q-1}}\;\left(a.e.\;s\in S\subseteq R^{n}\right). \end{array}$$

## 3. Main Results

In this study, we focus on the univariate PDFs on R, and we consider f(x) and g(x) defined on R as a special case of general f(s) and g(s) in Section 2.3. Hereafter, we refer to (Equation 10) and (Equation 12), as the equality condition of Hölder’s inequality and reverse Hölder’s inequality, respectively. Thanks to these equality conditions, we obtained our results systematically.

Let p(x) be a univariate PDF defined on R. Assume that p(x) is a measurable function which is integrable with respect to *x*. In addition, let B(·,·) denote the Beta function (cf. [42], p. 253). Then, we can form the following statements.

**Theorem** **1.**
*(Tsallis entropy maximization for 1<q<3): Suppose 1<q<3. $p_{\mathrm{opt}}\left(x\right)$, defined by*
$$\begin{array}{c} p_{\mathrm{opt}}\left(x\right)=\frac{1}{Z_{q}}\exp_{q}\left(-\beta_{q}x^{2}\right)\;\mathrm{with}\;Z_{q}=\sqrt{\frac{3-q}{q-1}}\sigma B\left(\frac{1}{q-1}-\frac{1}{2},\frac{1}{2}\right)\;\mathrm{and}\;\beta_{q}=\frac{1}{\left(3-q\right)\sigma^{2}}, \end{array}$$
*is the unique maximizer of the Tsallis entropy $H_{q}^{\mathrm{Tsallis}}\left[p\right]$ in ([Disp-formula FD2a-entropy-21-00549]) under the constraints $⟪p(x)⟫=1$ of (3b) and $⟪\left(x^{2}-\sigma^{2}\right)p^{q}\left(x\right)⟫=0$ of (3c) in*
**T2**
*.*


**Corollary** **1.**
*For q≥3, Tsallis entropy $H_{q}^{\mathrm{Tsallis}}\left[p\right]$ is bounded, but has no maximizer. Namely, there exist PDFs p(x), such that $H_{q}^{\mathrm{Tsallis}}\left[p\right]\to \frac{1}{q-1}\;\leq \frac{1}{2}$, in other words, $⟪p^{q}\left(x\right)⟫\to 0$. (The idea for constructing such PDFs is from Tsukada and Suyari [23], where they proved that*
**R1**
*for $q\leq \frac{1}{3}$ becomes unbounded, i.e., $⟪p^{q}\left(x\right)⟫\to +\infty$.)*


The proof of this theorem (and corollary) is given in Section 4.1. As mentioned in Section 1, the above statement itself has already appeared in [24]. However, our proof is quite different to the one in [24], in the sense that it does not require *generalized relative entropy*, and the maximizer is explicitly ‘specified’ (not ‘given ab initio’). Namely, reverse Hölder’s inequality aids in finding the optimal solution.

The outline of the proof is as follows. First, for 1<q<3, the maximization of the Tsallis entropy $H_{q}^{\mathrm{Tsallis}}\left[p\right]$, in other words, the minimization of $T_{q}\left[p;\lambda_{q}\right]$ in ([Disp-formula FD3a-entropy-21-00549]), is related to reverse Hölder’s inequality in (Equation 11). Second, we observe that $T_{q}\left[p;\lambda_{q}\right]$ has the lower bound through reverse Hölder’s inequality. Third, the minimizer $p_{\mathrm{opt}}\left(x\right)$ achieving this bound is explicitly and uniquely constructed from the equality condition (Equation 12):(13)$$\begin{array}{c} p_{\mathrm{opt}}\left(x\right)=A^{\frac{1}{q}}{\left[\lambda_{q,\mathrm{opt}}x^{2}+\left(1-\lambda_{q,\mathrm{opt}}\right)\sigma^{2}\right]}^{\frac{-1}{q-1}}, \end{array}$$
where $A={\left[\frac{3-q}{2}\sigma^{2}\right]}^{\frac{q}{q-1}}{\left[\sqrt{\frac{3-q}{q-1}}\sigma B\left(\frac{1}{q-1}-\frac{1}{2},\frac{1}{2}\right)\right]}^{-q}$ and $\lambda_{q,\mathrm{opt}}=\frac{1}{2}\left(q-1\right)$.

**Remark** **1.**
*Even if we assume the additional constraint: $p\in C_{q}\;\left(\subset D\right)$ in*
**T2**
*, the proof of this theorem (as well as of Theorems 2 and 3) remains the same, since we do not require the finiteness of $⟪x^{2}p^{q}\left(x\right)⟫$ and $⟪p^{q}\left(x\right)⟫$ (i.e., $p\in C_{q}$) in the proof.*


**Remark** **2.***Another simple proof for optimality of $p_{\mathrm{opt}}$ is given as follows. The idea is due to Moriguti’s argument (*cf. *[5], p. 288), where $p_{\mathrm{opt}}^{q}\left(x\right)$ and any $p^{q}\left(x\right)$ are directly related by the Taylor expansion for each x∈R:*(14)$$\begin{array}{c} p^{q}\left(x\right)=p_{\mathrm{opt}}^{q}\left(x\right)+qp_{\mathrm{opt}}^{q-1}\left(x\right)\left[p\left(x\right)-p_{\mathrm{opt}}\left(x\right)\right]+\frac{q(q-1)}{2}p_{\mathrm{int}}^{q-2}\left(x\right){\left[p\left(x\right)-p_{\mathrm{opt}}\left(x\right)\right]}^{2}, \end{array}$$*where $p_{\mathrm{int}}\left(x\right)\;\left(\geq 0\right)$ has a value between p(x) and $p_{\mathrm{opt}}\left(x\right)$. Substituting (Equation 13) into the second term of the right-hand side of (Equation 14), we have*(15)$$\begin{array}{cc} & \left[\lambda_{q,\mathrm{opt}}x^{2}+\left(1-\lambda_{q,\mathrm{opt}}\right)\sigma^{2}\right]\left[p^{q}\left(x\right)-p_{\mathrm{opt}}^{q}\left(x\right)\right]\\ = & qA^{\frac{q-1}{q}}\left[p\left(x\right)-p_{\mathrm{opt}}\left(x\right)\right]+\left[\lambda_{q,\mathrm{opt}}x^{2}+\left(1-\lambda_{q,\mathrm{opt}}\right)\sigma^{2}\right]\frac{q(q-1)}{2}p_{\mathrm{int}}^{q-2}\left(x\right){\left[p\left(x\right)-p_{\mathrm{opt}}\left(x\right)\right]}^{2}. \end{array}$$
*With the constraints (3b) and (3c): $⟪x^{2}p_{\mathrm{opt}}^{q}\left(x\right)⟫=\sigma^{2}⟪p_{\mathrm{opt}}^{q}\left(x\right)⟫$, $⟪x^{2}p^{q}\left(x\right)⟫=\sigma^{2}⟪p^{q}\left(x\right)⟫$, and $⟪p(x)⟫=⟪p_{\mathrm{opt}}\left(x\right)⟫=1$, integrating (Equation 15) over R, for any p(x), we find*
(16)$$\begin{array}{c} \sigma^{2}\left[⟪p^{q}\left(x\right)⟫-⟪p_{\mathrm{opt}}^{q}\left(x\right)⟫\right]=\frac{q(q-1)}{2}⟪\left[\lambda_{q,\mathrm{opt}}x^{2}+\left(1-\lambda_{q,\mathrm{opt}}\right)\sigma^{2}\right]p_{\mathrm{int}}^{q-2}\left(x\right){\left[p\left(x\right)-p_{\mathrm{opt}}\left(x\right)\right]}^{2}⟫\geq 0, \end{array}$$
*since $\lambda_{q,\mathrm{opt}}x^{2}+\left(1-\lambda_{q,\mathrm{opt}}\right)\sigma^{2}>0$ follows from $0<\lambda_{q,\mathrm{opt}}\left(=\frac{1}{2}\left(q-1\right)\right)<1$. Therefore, $p_{\mathrm{opt}}$ in (Equation 13) is a unique optimal solution to*
**T2**
*, as the equality holds only if $p=p_{\mathrm{opt}}$.*


**Theorem** **2.**
*(Tsallis entropy maximization for 0<q<1): Suppose 0<q<1. $p_{\mathrm{opt}}\left(x\right)$, as defined by*
$$\begin{array}{c} p_{\mathrm{opt}}\left(x\right)\;=\left\{\begin{array}{c} \frac{1}{Z_{q}}\exp_{q}\left(-\beta_{q}x^{2}\right)\;\;\left(x\in {\bar{S}}_{q,\mathrm{opt}}\right)\\ 0\;\;(x\in R\backslash {\bar{S}}_{q,\mathrm{opt}}), \end{array}\right.\\ \mathrm{with}\;\;\;{\bar{S}}_{q,\mathrm{opt}}=\left[-\sqrt{\frac{3-q}{1-q}}\sigma ,\sqrt{\frac{3-q}{1-q}}\sigma \right],\;Z_{q}=\sqrt{\frac{3-q}{1-q}}\sigma B\left(\frac{1}{1-q}+1,\frac{1}{2}\right),\;\mathrm{and}\;\;\;\beta_{q}=\frac{1}{\left(3-q\right)\sigma^{2}}, \end{array}$$
*is the unique maximizer of the Tsallis entropy $H_{q}^{\mathrm{Tsallis}}\left[p\right]$ in ([Disp-formula FD2a-entropy-21-00549]) under the constraints $⟪p(x)⟫=1$ of (3b) and $⟪\left(x^{2}-\sigma^{2}\right)p^{q}\left(x\right)⟫=0$ of (3c) in*
**T2**
*.*


The proof of this theorem is given in Section 4.2. In the case for 0<q<1, the maximization of $T_{q}\left[p;\lambda_{q}\right]$ is recast as Hölder’s inequality in (Equation 9), where, similar to the argument in the proof of Theorem 1, construction of $p_{\mathrm{opt}}$ and $\lambda_{q,\mathrm{opt}}$ and verification of its optimality are carried out simultaneously. The maximizer $p_{\mathrm{opt}}$ for 0<q<1 is uniquely determined from the equality condition (Equation 10):(17)$$\begin{array}{c} p_{\mathrm{opt}}\left(x\right)=\left\{\begin{array}{c} \frac{A}{B}{\left[\lambda_{q,\mathrm{opt}}x^{2}+\left(1-\lambda_{q,\mathrm{opt}}\right)\sigma^{2}\right]}^{\frac{1}{1-q}}\;\;\left(x\in {\bar{S}}_{q,\mathrm{opt}}\right)\\ 0\;\;\left(x\in R\backslash {\bar{S}}_{q,\mathrm{opt}}\right), \end{array}\right. \end{array}$$
where $A/B={\left[\frac{3-q}{2}\sigma^{2}\right]}^{\frac{1}{q-1}}{\left[\sqrt{\frac{3-q}{1-q}}\sigma B\left(\frac{1}{1-q}+1,\frac{1}{2}\right)\right]}^{-1}$ and $\lambda_{q,\mathrm{opt}}=\frac{1}{2}\left(q-1\right)\;\left(<0\right)$ are uniquely determined, and the associated ${\bar{S}}_{q,\mathrm{opt}}$ is uniquely determined as S¯q,opt=−(λq,opt−1λq,optσ,(λq,opt−1λq,optσ.

**Remark** **3.**
*Another simple proof for optimality of $p_{\mathrm{opt}}$ is given by following Moriguti’s argument ([5], p. 288). Similar to the case for 1<q<3 in Remark 2, (Equation 14) holds for $x\in {\bar{S}}_{q,\mathrm{opt}}\subset R$. As for $x\in R\backslash {\bar{S}}_{q,\mathrm{opt}}$, $p_{\mathrm{opt}}\left(x\right)=0$ from (Equation 17). We then have*
(18)$$\begin{array}{c} \left[\lambda_{q,\mathrm{opt}}x^{2}+\left(1-\lambda_{q,\mathrm{opt}}\right)\sigma^{2}\right]\left[p^{q}\left(x\right)-p_{\mathrm{opt}}^{q}\left(x\right)\right]\\ \;=\{\begin{array}{l} qA^{\frac{q-1}{q}}\left[p\left(x\right)-p_{\mathrm{opt}}\left(x\right)\right]+\left[\lambda_{q,\mathrm{opt}}x^{2}+\left(1-\lambda_{q,\mathrm{opt}}\right)\sigma^{2}\right]\frac{q(q-1)}{2}p_{\mathrm{int}}^{q-2}\left(x\right){\left[p\left(x\right)-p_{\mathrm{opt}}\left(x\right)\right]}^{2}\;\left(x\in {\bar{S}}_{q,\mathrm{opt}}\right)\\ {⟪\left[\lambda_{q,\mathrm{opt}}x^{2}+\left(1-\lambda_{q,\mathrm{opt}}\right)\sigma^{2}\right]p^{q}\left(x\right)⟫}_{R\backslash {\bar{S}}_{q,\mathrm{opt}}}\;\left(x\in R\backslash {\bar{S}}_{q,\mathrm{opt}}\right), \end{array} \end{array}$$
*where ${⟪\cdot ⟫}_{R\backslash {\bar{S}}_{q,\mathrm{opt}}}=\int_{R\backslash {\bar{S}}_{q,\mathrm{opt}}}\cdot \;dx$ is used for notational simplicity. Integrating (18) over R, for any p satisfying the constraints (3b) and (3c), we find*
(19)$$\begin{array}{cc} \; & \sigma^{2}\left[⟪p^{q}\left(x\right)⟫-⟪p_{\mathrm{opt}}^{q}\left(x\right)⟫\right]\\ \; & \;=qA^{\frac{q-1}{q}}{⟪p\left(x\right)-p_{\mathrm{opt}}\left(x\right)⟫}_{{\bar{S}}_{q,\mathrm{opt}}}+\frac{q(q-1)}{2}{⟪\left[\lambda_{q,\mathrm{opt}}x^{2}+\left(1-\lambda_{q,\mathrm{opt}}\right)\sigma^{2}\right]p_{\mathrm{int}}^{q-2}\left(x\right){\left[p\left(x\right)-p_{\mathrm{opt}}\left(x\right)\right]}^{2}⟫}_{{\bar{S}}_{q,\mathrm{opt}}}\\ \; & \;+{⟪\left[\lambda_{q,\mathrm{opt}}x^{2}+\left(1-\lambda_{q,\mathrm{opt}}\right)\sigma^{2}\right]p^{q}\left(x\right)⟫}_{R\backslash {\bar{S}}_{q,\mathrm{opt}}}\;\leq 0, \end{array}$$
*since the first term on the right-hand side ≤0 as ${⟪p(x)⟫}_{{\bar{S}}_{q,\mathrm{opt}}}\leq 1$, the second term ≤0 from the definition of ${\bar{S}}_{q,\mathrm{opt}}$ and the fact that 0<q<1, and the third term ≤0 because of the definition of ${\bar{S}}_{q,\mathrm{opt}}$. Since the equality in (Equation 19) holds only if $p=p_{\mathrm{opt}}$, this implies that $p_{\mathrm{opt}}$ in (Equation 17) is a unique optimal solution to*
**T2**
*.*


**Theorem** **3.**
*(Tsallis entropy maximization for q=0): Suppose q=0. $p_{\mathrm{opt}}\left(x\right)$, defined by*
(20)$$\begin{array}{c} p_{\mathrm{opt}}\left(x\right)\;=\left\{\begin{array}{c} \mathrm{arbitrary}\;\mathrm{positive}\;\mathrm{value}\;\;(x\in {\bar{S}}_{q,\mathrm{opt}}=\left[-\sqrt{3}\sigma ,\sqrt{3}\sigma \right])\\ 0\;\;(x\in R\backslash {\bar{S}}_{q,\mathrm{opt}}), \end{array}\right.\\ with\;\;{⟪p_{\mathrm{opt}}\left(x\right)⟫}_{{\bar{S}}_{q,\mathrm{opt}}}=1, \end{array}$$
*is the unique representation of the maximizer of the Tsallis entropy $H_{q}^{\mathrm{Tsallis}}\left[p\right]$ in ([Disp-formula FD2a-entropy-21-00549]) under the constraints $⟪p(x)⟫=1$ of (3b) and $⟪\left(x^{2}-\sigma^{2}\right)p^{0}\left(x\right)⟫=0$ of (3c) in*
**T2**
*.*


The proof of this theorem is given in Section 4.3, where the associated Hölder’s inequality is given as ${∥fg∥}_{1}\leq {∥f∥}_{\infty }{∥g∥}_{1}$, and we follow the arguments in the proof of Theorem 2 for 0<q<1. (This exceptional case (q=0) is also considered in **T1**, where the same result is obtained through a more direct graphical argument, after proving that any candidate $p_{*}\left(x\right)$ for the maximizer $p_{\mathrm{opt}}\left(x\right)$ is defined only on a simply connected interval $S_{*}$ that is symmetric about the origin O. The proof is straightforward but lengthy, so we omit it here.) However, as opposed to the case for 0<q<1, the equality condition is not available in the form of (Equation 10) for q=0, and we directly verify that
$$\begin{array}{c} f_{*}\left(x\right)\;\;=\;\left\{\begin{array}{c} \mathrm{sgn}\left[g\left(x\right)\right]\;\;\left(a.e.\;x\in {\bar{S}}_{q,\mathrm{opt}}\right)\\ 0\;\;\left(x\in R\backslash {\bar{S}}_{q,\mathrm{opt}}\right) \end{array}\right. \end{array}$$
is the unique solution satisfying the equality in (Equation 9), as shown in Lemma 1. Namely, for any feasible solutions p(x) satisfying the constraints (3b) and (3c), we find that $f_{*}\left(x\right)$ is associated with the unique maximizer $p_{\mathrm{opt}}\left(x\right)$ of $T_{q}\left[p;\lambda_{q}\right]$ from (Equation 52), and hence, $p_{\mathrm{opt}}\left(x\right)$ for q=0 is obtained as in (Equation 20).

**Remark** **4.**
*We note that the optimal solution shown in ([2], p. 2399, Figure 1) for q=0, which is obtained by setting q→0 in (Equation 17), is a special case of (Equation 20).*


**Theorem** **4.**
*(Rényi entropy maximization for q>1): Suppose q>1. $p_{\mathrm{opt}}\left(x\right)$, as defined by*
$$\begin{array}{c} p_{\mathrm{opt}}\left(x\right)\;=\left\{\begin{array}{c} \frac{1}{Z_{q}^{*}}\exp_{q^{*}}\left(-\beta_{q}^{*}x^{2}\right)\;\;\left(x\in {\underset{\bar{}}{S}}_{\mathrm{opt}}\right)\\ 0\;\;(x\in R\backslash {\underset{\bar{}}{S}}_{\mathrm{opt}}), \end{array}\right.\\ \mathrm{with}\;\;q_{*}=2-q,\;\;{\underset{\bar{}}{S}}_{\mathrm{opt}}=\left[-\sqrt{\frac{3q-1}{q-1}}\sigma ,\sqrt{\frac{3q-1}{q-1}}\sigma \right],\;Z_{q}^{*}=\sqrt{\frac{3q-1}{q-1}}\sigma B\left(\frac{q}{q-1},\frac{1}{2}\right),\;\mathrm{and}\;\;\;\beta_{q}^{*}=\frac{1}{\left(3q-1\right)\sigma^{2}}, \end{array}$$
*is the unique maximizer of Rényi entropy in ([Disp-formula FD4a-entropy-21-00549]) under the constraints $⟪p(x)⟫=1$ of (5b) (or (7b)) and $⟪x^{2}p\left(x\right)⟫=\sigma^{2}$ of (5c) (or (7c)).*


The proof of this theorem is given in Section 4.4. The minimization of $R_{q}\left[p;\lambda_{q}\right]$ in (7b) for q>1 is related to reverse Hölder’s inequality in (Equation 11). In contrast to those of Theorems 1–3, the proof of Theorem 4, which can be found in Section 4.4, follows from two steps. In the first step, we construct a candidate for the minimizer (i.e., $p_{\mathrm{opt}}\left(x\right)$, see (Equation 61) below), whose support becomes ${\underset{\bar{}}{S}}_{\mathrm{opt}}$, and we determine the associated $\lambda_{q,\mathrm{opt}}$ and ${\underset{\bar{}}{S}}_{\mathrm{opt}}$ through the equality condition of reverse Hölder’s inequality. In doing so, as shown in Figure 1, we introduce a subset of feasible solutions p(x), in other words, Q, which satisfies the constraints (5b) and (5c), and an additional constraint: $p^{q}\left(x\right)+\lambda_{q}\left(x^{2}-\sigma^{2}\right)p\left(x\right)>0\;\left(x\in R\right)$. In the second step, after obtaining a candidate $p_{\mathrm{opt}}\left(x\right)\in Q$, we verify that this $p_{\mathrm{opt}}\left(x\right)$ is indeed the unique minimizer of $R_{q}\left[p;\lambda_{q}\right]$ by directly comparing $⟪p_{\mathrm{opt}}^{q}\left(x\right)⟫$ and $⟪p^{q}\left(x\right)⟫$ for any feasible solutions p(x) satisfying the constraints (5b) and (5c).

**Remark** **5.**
*We note that the first proof for this optimality of $p_{\mathrm{opt}}$ has been given in Moriguti [5], in which the essential idea is the Taylor expansion shown in the argument below (18).*


**Theorem** **5.**
*(Rényi entropy maximization for $\frac{1}{3}<\;q<1$): Suppose $\frac{1}{3}<q<1$. $p_{\mathrm{opt}}\left(x\right)$, defined by*
$$\begin{array}{c} p_{\mathrm{opt}}\left(x\right)=\frac{1}{Z_{q}^{*}}\exp_{q^{*}}\left(-\beta_{q}^{*}x^{2}\right)\\ \mathrm{with}\;q_{*}=2-q,\;Z_{q}^{*}=\sqrt{\frac{3q-1}{1-q}}\sigma B\left(\frac{1}{1-q}-\frac{1}{2},\frac{1}{2}\right)\;\mathrm{and}\;\beta_{q}^{*}=\frac{1}{\left(3q-1\right)\sigma^{2}}, \end{array}$$
*is the unique maximizer of Rényi entropy ([Disp-formula FD4a-entropy-21-00549]) under the constraints (5b) and (5c) (or (7b) and (7c)).*


The proof of this theorem is given in Section 4.5. Maximization of $R_{q}\left[p;\lambda_{q}\right]$ is related to Hölder’s inequality in (Equation 9) and the proof follows two steps, similar to the proof for Theorem 4. In the first step, we construct a candidate for the maximizer (i.e., $p_{\mathrm{opt}}\left(x\right)$, given below by (Equation 72)) and determine $\lambda_{q,\mathrm{opt}}$ through the equality condition of Hölder’s inequality. In the second step, after obtaining a candidate $p_{\mathrm{opt}}$, we verify that this $p_{\mathrm{opt}}$ is indeed the unique maximizer of $R_{q}\left[p;\lambda_{q}\right]$ by directly comparing $⟪p_{\mathrm{opt}}^{q}\left(x\right)⟫$ and $⟪p^{q}\left(x\right)⟫$ for any feasible solutions p(x) satisfying the constraints (5b) and (5c). This verification is done as in the proofs for Theorem 4. Although omitted here, using essentially the same argument as in Remark 1, another simple proof based on Moriguti [5] is possible.

**Remark** **6.**
*Tsukada and Suyari [23] have proved that*
**R1**
*for $0<q\leq \frac{1}{3}$ becomes unbounded. As for the exceptional case of q=0, the upper and lower bounds of $⟪p^{q}\left(x\right)⟫\left(=⟪p^{0}\left(x\right)⟫\right)$ are argued as follows. First, if we consider the Gaussian distribution that satisfies (5b) and (5c), this gives us $⟪p^{0}\left(x\right)⟫={⟪1⟫}_{R}=\infty$, and it implies there is no maximizer. Next, consider a particular distribution given by*
(21)$$\begin{array}{c} \Delta \left(x\right)=\left\{\begin{array}{cc} \delta^{-1} & \left(x\in \left[\bar{\sigma },\bar{\sigma }+\delta \right]\right)\\ 0 & (otherwise), \end{array}\right. \end{array}$$
*with δ>0. This Δ(x) satisfies $⟪\Delta (x)⟫=\delta^{-1}\delta =1$ in (5b), and it also satisfies $⟪x^{2}\Delta \left(x\right)⟫=\sigma^{2}$ in (5c) when δ is arbitrary small, in other words,*
(22)$$\begin{array}{c} \forall \sigma ,\;\delta \;\left(\ll \sigma \right)\;\exists \bar{\sigma }\;⟪x^{2}\Delta \left(x\right)⟫={\bar{\sigma }}^{2}+\delta \bar{\sigma }+\frac{\delta^{2}}{3}=\sigma^{2}, \end{array}$$
*and, this particular distribution gives $⟪\Delta^{0}\left(x\right)⟫={⟪1⟫}_{\left[\bar{\sigma },\bar{\sigma }+\delta \right]}=\delta \to 0\;\left(\delta \to 0\right)$, which implies there is no minimizer. Therefore, problem*
**R1**
*(and*
**R2**
*) has no maximizer nor minimizer for q=0.*


## 4. Proof of Main Results

Following the outlines leading to Theorems 1–5 in Section 3, here we give their proofs.

### 4.1. Proof of Theorem 1

**Proof.** Let *p* be arbitrary feasible solutions to **T2** for 1<q<3, and let $p_{\mathrm{opt}}$ be its optimal solution, which is eventually constructed in (Equation 25). Let $\lambda_{q,\mathrm{opt}}$ be a particular value of the additional parameter $\lambda_{q}$ in **T2**, which is associated with $p_{\mathrm{opt}}$ and is eventually constructed in (Equation 29). Then, for any *p* and a particular $\lambda_{q,\mathrm{opt}}$ ($=\frac{1}{2}\left(q-1\right)$ in (Equation 29)), we define *f* and *g* as
(23a)$$\begin{array}{ccc} f(x) & = & p^{q}\left(x\right)\;\left(\;\geq 0\right), \end{array}$$
(23b)$$\begin{array}{ccc} g(x) & = & \lambda_{q,\mathrm{opt}}x^{2}+\left(1-\lambda_{q,\mathrm{opt}}\right)\sigma^{2}\;\;\left(>0\right). \end{array}$$First, we show $T_{q}\left[p;\lambda_{q}\right]$ is minimized in the following way:
(24a)$$\begin{array}{ccc} T_{q}\left[p;\lambda_{q}\right] & = & T_{q}\left[p;\lambda_{q,\mathrm{opt}}\right]=⟪fg⟫ \end{array}$$
(24b)$$\begin{array}{cc} = & {∥fg∥}_{1}\;\geq \;∥p^{q}{∥}_{\frac{1}{q}}{∥g∥}_{\frac{-1}{q-1}} \end{array}$$
(24c)$$\begin{array}{cc} = & {∥g∥}_{\frac{-1}{q-1}}\;\left(\mathrm{the}\;\mathrm{lower}\;\mathrm{bound}\right).\; \end{array}$$The first “=” in ([Disp-formula FD24a-entropy-21-00549]) follows from the fact that $T_{q}\left[p;\lambda_{q}\right]=\sigma^{2}⟪p^{q}\left(x\right)⟫+\lambda_{q}⟪\left(x^{2}-\sigma^{2}\right)p^{q}\left(x\right)⟫$ in ([Disp-formula FD3a-entropy-21-00549]) is independent from the value of $\lambda_{q}$, since any feasible solution *p* satisfies $⟪\left(x^{2}-\sigma^{2}\right)p^{q}\left(x\right)⟫=0$ in (3c), and the second “=” in ([Disp-formula FD24a-entropy-21-00549]) is immediate from (23). The “=” in (24b) follows from $p^{q}\left(x\right)\geq 0$ and g(x)>0 in (23b), since in (Equation 29) $\lambda_{q,\mathrm{opt}}=\frac{1}{2}\left(q-1\right)$, and it satisfies $0<\lambda_{q,\mathrm{opt}}<1$. The “≥” in (24b) follows from reverse Hölder’s inequality (Equation 11). The “=” in (24c) follows from $∥p^{q}{∥}_{\frac{1}{q}}={⟪\left|p(x)\right|⟫}^{q}=1$. (24c) implies that $T_{q}\left[p;\lambda_{q}\right]$ has the lower bound (i.e., the Tsallis entropy $H_{q}^{\mathrm{Tsallis}}\left[p\right]$ in (Equation 1) has the upper bound).Next, we construct a maximizer $p_{\mathrm{opt}}$ achieving this bound and show its uniqueness, which is done by checking the conditions where the “≥” in (24) become “=”; the only “≥” in (24b) becomes “=” if and only if the equality condition (Equation 12) is satisfied. Now, we rewrite the equality condition (Equation 12) (after assuming S=R in (Equation 12)) by using (23), which constructs (a candidate of) $p_{\mathrm{opt}}$:
(25)$$\begin{array}{c} p_{\mathrm{opt}}\left(x\right)\;=\;A^{\frac{1}{q}}{\left[\lambda_{q,\mathrm{opt}}x^{2}+\left(1-\lambda_{q,\mathrm{opt}}\right)\sigma^{2}\right]}^{\frac{-1}{q-1}}=\frac{1}{Z_{q}}\exp_{q}\left(-\beta_{q}x^{2}\right), \end{array}$$
where *A* and $\lambda_{q,\mathrm{opt}}\;\left(=\frac{1}{2}\left(q-1\right)\right)$ are uniquely determined, and hence, $Z_{q}$ and $\beta_{q}$ are also uniquely determined, as shown in the following calculations. We note the formula ([42], p. 253):
$$\begin{array}{c} \int_{0}^{\infty }\frac{dx}{x^{\alpha }{\left(1+x^{\gamma }\right)}^{\beta }}=\frac{1}{\gamma }B\left(\beta -\frac{1-\alpha }{\gamma },\;\frac{1-\alpha }{\gamma }\right) \end{array}$$
is repeatedly used for the integrations below. (More precisely, for (Equation 26), we set α=0(<1), $\beta =\frac{1}{q-1}\;\left(>0\right)$, γ=2(>0), and βγ>1−α is satisfied. For (27b), we set α=−2, 0(<1), $\beta =\frac{q}{q-1}\;\left(>0\right)$, γ=2(>0), and βγ>1−α is satisfied.)First, by substituting $p_{\mathrm{opt}}$ in (Equation 25) into the constraint (3b), and for 1<q<3 (finiteness of the left-hand side of (3b) requires the condition q<3), we have
(26)$$\begin{array}{c} ⟪p_{\mathrm{opt}}\left(x\right)⟫=\;A^{\frac{1}{q}}{\left[\left(1-\lambda_{q,\mathrm{opt}}\right)\sigma^{2}\right]}^{\frac{1}{1-q}}\sqrt{\frac{1-\lambda_{q,\mathrm{opt}}}{\lambda_{q,\mathrm{opt}}}}\sigma B\left(\frac{1}{q-1}-\frac{1}{2},\frac{1}{2}\right)=1. \end{array}$$On the other hand, substituting (Equation 25) into $n⟪p^{q}\left(x\right)⟫$ and $⟪x^{2}p^{q}\left(x\right)⟫$ in the constraint (3c), we have
(27a)$$\begin{array}{ccc} ⟪p_{\mathrm{opt}}^{q}\left(x\right)⟫ & = & A{\left[\left(1-\lambda_{q,\mathrm{opt}}\right)\sigma^{2}\right]}^{\frac{q}{1-q}}\sigma \left[\sqrt{\frac{1-\lambda_{q,\mathrm{opt}}}{\lambda_{q,\mathrm{opt}}}}B\left(\frac{1}{q-1}+\frac{1}{2},\frac{1}{2}\right)\right], \end{array}$$
(27b)$$\begin{array}{ccc} ⟪x^{2}p_{\mathrm{opt}}^{q}\left(x\right)⟫ & = & A{\left[\left(1-\lambda_{q,\mathrm{opt}}\right)\sigma^{2}\right]}^{\frac{q}{1-q}}\sigma^{3}\left[{\left(\frac{1-\lambda_{q,\mathrm{opt}}}{\lambda_{q,\mathrm{opt}}}\right)}^{\frac{3}{2}}B\left(\frac{1}{q-1}-\frac{1}{2},\frac{3}{2}\right)\right], \end{array}$$
and substitution of ([Disp-formula FD27a-entropy-21-00549]) and (27b) into (3c) yields
(28)$$\begin{array}{c} A{\left[\left(1-\lambda_{q,\mathrm{opt}}\right)\sigma^{2}\right]}^{\frac{q}{1-q}}\sigma^{3}\left[{\left(\frac{1-\lambda_{q,\mathrm{opt}}}{\lambda_{q,\mathrm{opt}}}\right)}^{\frac{3}{2}}B\left(\frac{1}{q-1}-\frac{1}{2},\frac{3}{2}\right)-\sqrt{\frac{1-\lambda_{q,\mathrm{opt}}}{\lambda_{q,\mathrm{opt}}}}B\left(\frac{1}{q-1}+\frac{1}{2},\frac{1}{2}\right)\right]=0. \end{array}$$Then, using the formula $B\left(x,y+1\right)=\frac{y}{x}B\left(x+1,y\right)$ ([42], p. 254) in (Equation 28), $\lambda_{q,\mathrm{opt}}$ is uniquely determined as
(29)$$\begin{array}{c} \frac{\lambda_{q,\mathrm{opt}}}{1-\lambda_{q,\mathrm{opt}}}=\frac{q-1}{3-q},\;i.e.,\;\lambda_{q,\mathrm{opt}}=\frac{1}{2}\left(q-1\right), \end{array}$$
and from (Equation 26) and (Equation 29) *A* is uniquely determined as
(30)$$\begin{array}{c} A={\left[\frac{3-q}{2}\sigma^{2}\right]}^{\frac{q}{q-1}}{\left[\sqrt{\frac{3-q}{q-1}}\sigma B\left(\frac{1}{q-1}-\frac{1}{2},\frac{1}{2}\right)\right]}^{-q}. \end{array}$$In (Equation 25), equating the second term to the third term, $\beta_{q}$ is uniquely determined as
$$\begin{array}{c} \beta_{q}=\frac{\lambda_{q,\mathrm{opt}}}{1-\lambda_{q,\mathrm{opt}}}{\left(q-1\right)}^{-1}\sigma^{-2}=\frac{1}{\left(3-q\right)\sigma^{2}}, \end{array}$$
and from (Equation 29) and (Equation 30), $Z_{q}$ is uniquely determined as
$$\begin{array}{c} Z_{q}={\left\{A^{\frac{1}{q}}{\left[\left(1-\lambda_{q,\mathrm{opt}}\right)\sigma^{2}\right]}^{\frac{1}{1-q}}\right\}}^{-1}=\sqrt{\frac{3-q}{q-1}}\sigma B\left(\frac{1}{q-1}-\frac{1}{2},\frac{1}{2}\right). \end{array}$$This proves that $p_{\mathrm{opt}}$ in (Equation 25) is a unique minimizer to **T2** for 1<q<3.Finally, we prove the Corollary to Theorem 1 in Section 3. For q≥3, let $p_{*}$ be the distribution satisfying (3b) and (3c), defined as
$$\begin{array}{c} p_{*}\left(x\right)=Z^{-1}{\left(|x|+\alpha \right)}^{-(1+\varepsilon )}, \end{array}$$
where a normalization factor $Z\;=\;\int {\left(|x|+\alpha \right)}^{-(1+\varepsilon )}dx$ and α,ε>0. What we are going to prove is $⟪p_{*}^{q}\left(x\right)⟫\to 0\;\;\left(\varepsilon \to 0\right)$, which is done as follows. First, straightforward integrations yield:
(31a)$$\begin{array}{ccc} Z & = & 2\alpha^{-\varepsilon }\varepsilon^{-1}, \end{array}$$
(31b)$$\begin{array}{ccc} ⟪p_{*}^{q}\left(x\right)⟫ & = & 2\alpha^{1-\bar{q}}{\left(\bar{q}-1\right)}^{-1}Z^{-q}, \end{array}$$
(31c)$$\begin{array}{ccc} ⟪x^{2}p_{*}^{q}\left(x\right)⟫ & = & 4\alpha^{3-\bar{q}}{\left(\bar{q}-1\right)}^{-1}{\left(\bar{q}-2\right)}^{-1}{\left(\bar{q}-3\right)}^{-1}Z^{-q}, \end{array}$$
where $\bar{q}\;=\;\left(1+\varepsilon \right)q\;>3$. Second, from the constraint in (3c), $2{\left(\bar{q}-2\right)}^{-1}{\left(\bar{q}-3\right)}^{-1}\alpha^{2}=\sigma^{2}$ is obtained, and this shows that α becomes finite and is determined by ε, *q*, and σ. Finally, substituting ([Disp-formula FD31a-entropy-21-00549]) into (31b), we obtain
(32)$$\begin{array}{c} ⟪p_{*}^{q}\left(x\right)⟫=2^{1-q}{\left(\bar{q}-1\right)}^{-1}\varepsilon^{q}, \end{array}$$
and hence $⟪p_{*}^{q}\left(x\right)⟫\to 0\;\;\left(\varepsilon \to 0\right)$, since for q≥3 in (Equation 32) ${\left(\bar{q}-1\right)}^{-1}\to {\left(q-1\right)}^{-1}$ and $\varepsilon^{q}\to 0\;\;\left(\varepsilon \to 0\right)$ in (Equation 32). □

### 4.2. Proof of Theorem 2

**Proof.** Let *p* be arbitrary feasible solutions to **T2** for 0<q<1, and let $p_{\mathrm{opt}}$ be its optimal solution, which is eventually constructed in (Equation 38). Let $\lambda_{q,\mathrm{opt}}$ be a particular value of the additional parameter $\lambda_{q}$ in **T2**, which is associated with $p_{\mathrm{opt}}$ and is eventually constructed in (Equation 42). Then, for any *p* and a particular $\lambda_{q,\mathrm{opt}}$ ($=\frac{1}{2}\left(q-1\right)$ in (Equation 42)), we define *f* and *g* as
(33a)$$\begin{array}{ccc} f(x) & = & p^{q}\left(x\right), \end{array}$$
(33b)$$\begin{array}{ccc} g(x) & = & \lambda_{q,\mathrm{opt}}x^{2}+\left(1-\lambda_{q,\mathrm{opt}}\right)\sigma^{2}, \end{array}$$
and we define an interval
S¯q,opt=−(λq,opt−1λq,optσ,(λq,opt−1λq,optσ.First, we show that $T_{q}\left[p;\lambda_{q}\right]$ is maximized in the following way:
(34a)$$\begin{array}{ccc} T_{q}\left[p;\lambda_{q}\right] & = & T_{q}\left[p;\lambda_{q,\mathrm{opt}}\right]=⟪fg⟫ \end{array}$$
(34b)$$\begin{array}{cc} \leq & {⟪fg⟫}_{{\bar{S}}_{q,\mathrm{opt}}}={⟪\left|fg\right|⟫}_{{\bar{S}}_{q,\mathrm{opt}}} \end{array}$$
(34c)$$\begin{array}{cc} = & {∥fg∥}_{1,{\bar{S}}_{q,\mathrm{opt}}}\leq {∥f∥}_{\frac{1}{q},{\bar{S}}_{q,\mathrm{opt}}}\;{∥g∥}_{\frac{1}{1-q},{\bar{S}}_{q,\mathrm{opt}}} \end{array}$$
(34d)$$\begin{array}{cc} \leq & {∥g∥}_{\frac{1}{1-q},{\bar{S}}_{q,\mathrm{opt}}}\;\;(\mathrm{the}\;\mathrm{upper}\;\mathrm{bound}),\; \end{array}$$
where $⟪\cdot ⟫=\int_{-\infty }^{\infty }\cdot \;dx$, ${⟪\cdot ⟫}_{{\bar{S}}_{q,\mathrm{opt}}}=\int_{{\bar{S}}_{q,\mathrm{opt}}}\cdot \;dx$, and ${∥\cdot ∥}_{\alpha ,{\bar{S}}_{q,\mathrm{opt}}}={\left(\int_{\;{\bar{S}}_{q,\mathrm{opt}}}{\left|\cdot \right|}^{\alpha }dx\right)}^{\frac{1}{\alpha }}$. The first “=” in ([Disp-formula FD34a-entropy-21-00549]) follows from the fact that $T_{q}\left[p;\lambda_{q}\right]=\sigma^{2}⟪p^{q}\left(x\right)⟫+\lambda_{q}⟪\left(x^{2}-\sigma^{2}\right)p^{q}\left(x\right)⟫$ in ([Disp-formula FD3a-entropy-21-00549]) is independent from the value of $\lambda_{q}$, since any feasible solution *p* satisfies $⟪\left(x^{2}-\sigma^{2}\right)p^{q}\left(x\right)⟫=0$ in (3c), and the second “=” in ([Disp-formula FD34a-entropy-21-00549]) is immediate from (33). The “≤” in (34b) is obtained from the following observation. By plotting the graph of g(x) for (any negative) $\lambda_{q,\mathrm{opt}}$, we observe that ${\bar{S}}_{q,\mathrm{opt}}$ is the set of *x* on which g(x) becomes positive. For any *f* and *g* in ([Disp-formula FD34a-entropy-21-00549]), we also observe from this graph that ${⟪fg⟫}_{{\bar{S}}_{q,\mathrm{opt}}}>{⟪fg⟫}_{S_{*}}$ for any set $S_{*}\;\left(\subset R\right)$ but ${\bar{S}}_{q,\mathrm{opt}}$, since
(35)$$\begin{array}{c} f\left(x\right)\geq 0\;\left(\forall x\in R\right),\;\;g\left(x\right)\geq 0\;\left(x\in {\bar{S}}_{q,\mathrm{opt}}\right),\;\mathrm{and}\;\;g\left(x\right)<0\;\left(x\in R\backslash {\bar{S}}_{q,\mathrm{opt}}\right). \end{array}$$On the other hand, the “=” in (34b) is immediate from $f\left(x\right)g\left(x\right)\geq 0\;\left(\forall x\in {\bar{S}}_{q,\mathrm{opt}}\right)$. The first “=” in (34c) follows from the definition of ${∥\cdot ∥}_{1,{\bar{S}}_{q,\mathrm{opt}}}$, and the inequality in (34c) follows from Hölder’s inequality (Equation 9). In view of ([Disp-formula FD33a-entropy-21-00549]), the final “≤” in (34d) follows from
$$\begin{array}{c} {∥f∥}_{\frac{1}{q},{\bar{S}}_{q,\mathrm{opt}}}={\left(\int_{{\bar{S}}_{q,\mathrm{opt}}}\left|p\left(x\right)\right|\;dx\right)}^{q}\leq {\left(\int_{-\infty }^{\infty }\left|p\left(x\right)\right|\;dx\right)}^{q}=1, \end{array}$$
and the resulting ${∥g∥}_{\frac{1}{1-q},{\bar{S}}_{q,\mathrm{opt}}}$ implies the upper bound of $T_{q}\left[p;\lambda_{q}\right]$ if $\lambda_{q,\mathrm{opt}}$ exists for a given *q* and σ.Next, we construct a maximizer $p_{\mathrm{opt}}$ achieving this bound and show its uniqueness, which is done by checking the conditions where all three “≤” in (34) become “=”. As for the “≤” in (34b), it becomes “=” if and only if p(x) becomes positive only in ${\bar{S}}_{q,\mathrm{opt}}$. Namely,
(36)$$\begin{array}{c} f\left(x\right)=p^{q}\left(x\right)\geq 0\;\;\left(x\in {\bar{S}}_{q,\mathrm{opt}}\right)\;\mathrm{and}\;f\left(x\right)=0\;\;\left(a.e.\;x\in R\backslash {\bar{S}}_{q,\mathrm{opt}}\right). \end{array}$$In other words, the “≤” in (34b) becomes “<” if the above condition (Equation 36) is violated, which is easily verified from the graph of g(x) and the above argument for the “≤” in (34b). On the other hand, in (34c), the “≤” becomes “=” if and only if the equality condition (Equation 10) is satisfied for $x\in {\bar{S}}_{q,\mathrm{opt}}$, in other words,
(37)$$\begin{array}{c} Ap\left(x\right)=B{\left|\lambda_{q,\mathrm{opt}}x^{2}+\left(1-\lambda_{q,\mathrm{opt}}\right)\sigma^{2}\right|}^{\frac{1}{1-q}}. \end{array}$$If *p* satisfies (Equation 36), it is immediate that A≠0 and B≠0 in (Equation 37), since ${\left[\lambda_{q,\mathrm{opt}}x^{2}+\left(1-\lambda_{q,\mathrm{opt}}\right)\sigma^{2}\right]}^{\frac{1}{1-q}}\geq 0\;\;\left(x\in {\bar{S}}_{q,\mathrm{opt}}\right)$, and *A* and *B* are not both 0 (cf. [40], p. 140). The conclusion is that, if these two conditions (Equation 36) and (Equation 37) are satisfied, a maximizer $p_{\mathrm{opt}}$ achieving the upper bound of $T_{q}\left[p;\lambda \right]$ is uniquely determined:
(38)$$\begin{array}{c} p_{\mathrm{opt}}\left(x\right)=\left\{\begin{array}{c} \frac{A}{B}{\left[\lambda_{q,\mathrm{opt}}x^{2}+\left(1-\lambda_{q,\mathrm{opt}}\right)\sigma^{2}\right]}^{\frac{1}{1-q}}=\frac{1}{Z_{q}}\exp_{q}\left(-\beta_{q}x^{2}\right)\;\;\left(x\in {\bar{S}}_{q,\mathrm{opt}}\right)\\ 0\;\;\left(x\in R\backslash {\bar{S}}_{q,\mathrm{opt}}\right), \end{array}\right. \end{array}$$
in which A/B, $\lambda_{q,\mathrm{opt}}$, $Z_{q}=\sqrt{\frac{3-q}{1-q}}\sigma B\left(\frac{1}{1-q}+1,\frac{1}{2}\right)$, $\beta_{q}=\frac{1}{\left(3-q\right)\sigma^{2}}$, and ${\bar{S}}_{q,\mathrm{opt}}=\left[-\sqrt{\frac{3-q}{1-q}}\sigma ,\sqrt{\frac{3-q}{1-q}}\sigma \right]$ are uniquely determined, as shown in the following calculations. We note the formula ([42], p. 253):
$$\begin{array}{c} \int_{0}^{1}x^{\alpha }{\left(1-x^{\gamma }\right)}^{\beta }dx=\frac{1}{\gamma }B\left(1+\beta ,\;\frac{1+\alpha }{\gamma }\right) \end{array}$$
is repeatedly used for the integrations below. (More precisely, for (Equation 39), we set α=0(>−1), $\beta =\frac{1}{1-q}\;\left(>-1\right)$, and γ=2(>0). For (40b), we set α=2, 0(>−1), $\beta =\frac{q}{1-q}\;\left(>-1\right)$, and γ=2(>0).)By substituting $p_{\mathrm{opt}}\left(x\right)$ in (Equation 38) into the constraint (3b), we have
(39)$$\begin{array}{c} ⟪p_{\mathrm{opt}}\left(x\right)⟫=\;\frac{A}{B}{\left[\left(1-\lambda_{q,\mathrm{opt}}\right)\sigma^{2}\right]}^{\frac{1}{1-q}}\sqrt{\frac{\lambda_{q,\mathrm{opt}}-1}{\lambda_{q,\mathrm{opt}}}}\sigma B\left(\frac{1}{1-q}+1,\frac{1}{2}\right)=1. \end{array}$$On the other hand, by substituting (Equation 38) into $⟪p^{q}\left(x\right)⟫$ and $⟪x^{2}p^{q}\left(x\right)⟫$ in the constraint (3c), we have
(40a)$$\begin{array}{ccc} ⟪p_{\mathrm{opt}}^{q}\left(x\right)⟫ & = & {\left(\frac{A}{B}\right)}^{q}{\left[\left(1-\lambda_{q,\mathrm{opt}}\right)\sigma^{2}\right]}^{\frac{q}{1-q}}\sigma \left[\sqrt{\frac{\lambda_{q,\mathrm{opt}}-1}{\lambda_{q,\mathrm{opt}}}}B\left(\frac{1}{1-q},\frac{1}{2}\right)\right], \end{array}$$
(40b)$$\begin{array}{ccc} ⟪x^{2}p_{\mathrm{opt}}^{q}\left(x\right)⟫ & = & {\left(\frac{A}{B}\right)}^{q}{\left[\left(1-\lambda_{q,\mathrm{opt}}\right)\sigma^{2}\right]}^{\frac{q}{1-q}}\sigma^{3}\left[{\left(\frac{\lambda_{q,\mathrm{opt}}-1}{\lambda_{q,\mathrm{opt}}}\right)}^{\frac{3}{2}}B\left(\frac{1}{1-q},\frac{3}{2}\right)\right], \end{array}$$
and substitution of ([Disp-formula FD40a-entropy-21-00549]) and (40b) into (3c) yields
(41)$$\begin{array}{c} {\left(\frac{A}{B}\right)}^{q}{\left[\left(1-\lambda_{q,\mathrm{opt}}\right)\sigma^{2}\right]}^{\frac{1}{1-q}}\sigma^{3}\left[{\left(\frac{\lambda_{q,\mathrm{opt}}-1}{\lambda_{q,\mathrm{opt}}}\right)}^{\frac{3}{2}}B\left(\frac{1}{1-q},\frac{3}{2}\right)-\sqrt{\frac{\lambda_{q,\mathrm{opt}}-1}{\lambda_{q,\mathrm{opt}}}}B\left(\frac{1}{1-q},\frac{1}{2}\right)\right]=0. \end{array}$$Then, using the formula $B\left(x,y+1\right)=\frac{y}{x+y}B\left(x,y\right)$ ([42], p. 254) in (Equation 41), $\lambda_{q,\mathrm{opt}}$ is uniquely determined as
(42)$$\begin{array}{c} \frac{\lambda_{q,\mathrm{opt}}}{\lambda_{q,\mathrm{opt}}-1}=\frac{1-q}{3-q},\;i.e.,\;\lambda_{q,\mathrm{opt}}=\frac{1}{2}\left(q-1\right) \end{array}$$
and from (Equation 39) and (Equation 42) A/B is uniquely determined as
$$\begin{array}{c} \frac{A}{B}={\left[\frac{3-q}{2}\sigma^{2}\right]}^{\frac{1}{q-1}}{\left[\sqrt{\frac{3-q}{1-q}}\sigma B\left(\frac{1}{1-q}+1,\frac{1}{2}\right)\right]}^{-1}. \end{array}$$In (Equation 38), equating the second term to the third term, $\beta_{q}$ is uniquely determined as
(43)$$\begin{array}{c} \beta_{q}=\frac{\lambda_{q,\mathrm{opt}}}{1-\lambda_{q,\mathrm{opt}}}{\left(q-1\right)}^{-1}\sigma^{-2}=\frac{1}{\left(3-q\right)\sigma^{2}}, \end{array}$$
and from (Equation 29) and (Equation 30), $Z_{q}$ is uniquely determined as
(44)$$\begin{array}{c} Z_{q}={\left\{\frac{A}{B}{\left[\left(1-\lambda_{q,\mathrm{opt}}\right)\sigma^{2}\right]}^{\frac{1}{1-q}}\right\}}^{-1}=\sqrt{\frac{3-q}{1-q}}\sigma B\left(\frac{1}{1-q}+1,\frac{1}{2}\right). \end{array}$$Thus, from (Equation 43), and (Equation 44), $p_{\mathrm{opt}}$ is uniquely obtained as in (Equation 38).To see that $p_{\mathrm{opt}}$ makes all “≤” in (34) “=”, finally, we check the last “≤” in (34d) becomes “=”, which is immediate since in (34c) ${∥f∥}_{\frac{1}{q},{\bar{S}}_{q,\mathrm{opt}}}=\int_{{\bar{S}}_{q,\mathrm{opt}}}p_{\mathrm{opt}}\left(x\right)dx=1$. Therefore, it is concluded that $T_{q}\left[p;\lambda_{q}\right]$ is uniquely maximized by $p_{\mathrm{opt}}$ in (Equation 38) for 0<q<1. □

### 4.3. Proof of Lemma 1 and Theorem 3

Let *p* be arbitrary feasible solutions to **T2** for q=0, and let $p_{\mathrm{opt}}$ be its optimal solution, which is eventually constructed in (Equation 53). Let $\lambda_{q,\mathrm{opt}}$ be a particular value of the additional parameter $\lambda_{q}$ in **T2**, which is associated with $p_{\mathrm{opt}}$ and is eventually constructed in (Equation 54). Then, for any *p* and a particular $\lambda_{q,\mathrm{opt}}$ ($=-\frac{1}{2}$ in (Equation 54)), we define *f* and *g* as
(45a)$$\begin{array}{ccc} f(x) & = & p^{0}\left(x\right)=\left\{\begin{array}{c} 1\;\;\mathrm{if}\;\;0<p(x)<\infty \\ 0\;\;\mathrm{if}\;\;p(x)=0, \end{array}\right. \end{array}$$
(45b)$$\begin{array}{ccc} g(x) & = & \lambda_{q,\mathrm{opt}}x^{2}+\left(1-\lambda_{q,\mathrm{opt}}\right)\sigma^{2}, \end{array}$$
and we define an interval
(46)S¯q,opt=−(λq,opt−1λq,optσ,(λq,opt−1λq,optσ.

In ([Disp-formula FD45a-entropy-21-00549]), as a convention, we take $0^{0}=0$, and $p\left(x\right)<\infty \;\left(a.e.\;x\in {\bar{S}}_{q,\mathrm{opt}}\right)$. Then, ${∥f∥}_{\infty }=1$ follows from ([Disp-formula FD45a-entropy-21-00549]). Now, we define $f_{*}$ as
(47)$$\begin{array}{c} f_{*}\left(x\right)\;\;=\;\left\{\begin{array}{c} \mathrm{sgn}\left[g\left(x\right)\right]\;\;\left(a.e.\;x\in {\bar{S}}_{q,\mathrm{opt}}\right)\\ 0\;\;\left(x\in R\backslash {\bar{S}}_{q,\mathrm{opt}}\right). \end{array}\right. \end{array}$$

We note this particular $f_{*}\left(x\right)=\mathrm{sgn}\left[g\left(x\right)\right]$ is proved to be the unique maximizer of ${∥fg∥}_{1,{\bar{S}}_{q,\mathrm{opt}}}$ in the following Lemma 1 (as a minor modification of Lemma 4 in [35]).

**Lemma** **1.**
*(cf. Lemma 4 in [35]). Let S be an arbitrary subset in R, with μ(S)>0. For $f\in L^{\infty }\left(S\right)$ and $g\in L^{1}\left(S\right)$, assume g(x)≠0,a.e.onS. Then, $f_{*}\left(x\right)=\mathrm{sgn}\left[g\left(x\right)\right]\;\;(a.e.\;x\in S)$ is the unique maximizer of the functional ${∥fg∥}_{1,S}$ in (Equation 9).*


**Proof of** **Lemma 1.**First, thanks to Hölder’s inequality, see (Equation 9), ${∥fg∥}_{1,S}$ is maximized by $f_{*}$, since
$$\begin{array}{c} ∥f_{*}{\;g∥}_{1,S}={⟪\mathrm{sgn}[g(x)]\;\;g(x)⟫}_{S}={⟪|g(x)|t⟫}_{S}={∥g∥}_{1,S}. \end{array}$$Second, the unique representation of this maximizer $f_{*}$ is shown by proof by contradiction, as follows. Suppose another maximizer ${\bar{f}}_{*}$ exists and it maximizes ${∥fg∥}_{1,S}$, in other words, $∥{\bar{f}}_{*}{\;g∥}_{1,S}={∥g∥}_{1,S}$. Then, for any given $g\in L^{1}\left(S\right)$, the following is satisfied:
(48)$$\begin{array}{c} ∥f_{*}{\;g∥}_{1,S}-{∥{\bar{f}}_{*}\;g∥}_{1,S}=0. \end{array}$$Now, using the identities $f_{*}\left(x\right)g\left(x\right)=\mathrm{sgn}\left[g\left(x\right)\right]\cdot g\left(x\right)\geq 0$ and $\left|g\left(x\right)\right|=f_{*}\left(x\right)g\left(x\right)$, we obtain $|f_{*}\left(x\right)g\left(x\right)|-|{\bar{f}}_{*}\left(x\right)g\left(x\right)|=f_{*}\left(x\right)g\left(x\right)-|{\bar{f}}_{*}\left(x\right)||g\left(x\right)|$ and $f_{*}\left(x\right)g\left(x\right)-|{\bar{f}}_{*}\left(x\right)||g\left(x\right)|=f_{*}\left(x\right)g\left(x\right)-|{\bar{f}}_{*}\left(x\right|f_{*}\left(x\right)g\left(x\right)$, respectively, resulting in the equality
(49)$$\begin{array}{c} |f_{*}\left(x\right)g\left(x\right)|-|{\bar{f}}_{*}\left(x\right)g\left(x\right)|=f_{*}\left(x\right)g\left(x\right)-\left|{\bar{f}}_{*}\left(x\right)\right|f_{*}\left(x\right)g\left(x\right). \end{array}$$Substituting (Equation 49) into the left-hand side of (Equation 48) and using $\left|g\left(x\right)\right|=f_{*}\left(x\right)g\left(x\right)$, (Equation 48) is rewritten as
(50)$$\begin{array}{c} {⟪\left(1-|{\bar{f}}_{*}\left(x\right)|\right)f_{*}\left(x\right)\;g\left(x\right)⟫}_{S}={⟪\left(1-|{\bar{f}}_{*}\left(x\right)|)|g\left(x\right)\right|⟫}_{S}=0. \end{array}$$Now, keeping $0\leq {\bar{f}}_{*}\left(x\right)\leq 1$ and the assumption that g(x)≠0,a.e.onS in mind, (Equation 50) implies
$$\begin{array}{c} |{\bar{f}}_{*}\left(x\right)|=1,\;\mathrm{or}\;\mathrm{equivalently}\\ {\bar{f}}_{*}\left(x\right)=\sigma \left(x\right),\;\;a.e.\;\mathrm{on}\;S, \end{array}$$
where σ takes either −1 or 1. However, among such functions ${\bar{f}}_{*}$ having either −1 or 1 values, it is clear that $\mathrm{sgn}\left[g\left(x\right)\right]\;\left(=f_{*}\right)$ is the only one that makes ${⟪fg⟫}_{S}$ maximal. Thus, no ${\bar{f}}_{*}$ can exist except for $f_{*}$, and the uniqueness of the maximizer $f_{*}$ is verified. □

**Proof of** **Theorem 3.**First, we show that $T_{q}\left[p;\lambda_{q}\right]$ is maximized in the following way:
(51a)$$\begin{array}{ccc} T_{q}\left[p;\lambda_{q}\right] & = & T_{q}\left[p;\lambda_{q,\mathrm{opt}}\right]=⟪fg⟫ \end{array}$$
(51b)$$\begin{array}{cc} \leq & {⟪fg⟫}_{{\bar{S}}_{q,\mathrm{opt}}}={⟪\left|fg\right|⟫}_{{\bar{S}}_{q,\mathrm{opt}}} \end{array}$$
(51c)$$\begin{array}{cc} = & {∥fg∥}_{1,{\bar{S}}_{q,\mathrm{opt}}}\leq {∥f∥}_{\infty ,{\bar{S}}_{q,\mathrm{opt}}}{∥g∥}_{1,{\bar{S}}_{q,\mathrm{opt}}} \end{array}$$
(51d)$$\begin{array}{cc} \leq & {∥g∥}_{1,{\bar{S}}_{q,\mathrm{opt}}}\;\;(\mathrm{the}\;\mathrm{upper}\;\mathrm{bound}), \end{array}$$
where $⟪\cdot ⟫=\int_{-\infty }^{\infty }\cdot \;dx$, ${⟪\cdot ⟫}_{{\bar{S}}_{q,\mathrm{opt}}}=\int_{{\bar{S}}_{q,\mathrm{opt}}}\cdot \;dx$, and ${∥\cdot ∥}_{1,{\bar{S}}_{q,\mathrm{opt}}}=\int_{\;{\bar{S}}_{q,\mathrm{opt}}}\left|\cdot \right|dx$, and ${∥f∥}_{\infty ,{\bar{S}}_{q,\mathrm{opt}}}$ is the infinity norm of $f\left(x\right)\;\left(x\in {\bar{S}}_{q,\mathrm{opt}}\right)$, in other words, the essential supremum of $|f\left(x\right)|\;\left(x\in {\bar{S}}_{q,\mathrm{opt}}\right)$. The first “=” in ([Disp-formula FD51a-entropy-21-00549]) follows from the fact that $T_{q}\left[p;\lambda_{q}\right]=\sigma^{2}⟪p^{q}\left(x\right)⟫+\lambda_{q}⟪\left(x^{2}-\sigma^{2}\right)p^{q}\left(x\right)⟫$ in ([Disp-formula FD3a-entropy-21-00549]) is independent from the value of $\lambda_{q}$, since any feasible solution p(x) satisfies $⟪\left(x^{2}-\sigma^{2}\right)p^{q}\left(x\right)⟫=0$ in (3c), and the second “=” in ([Disp-formula FD51a-entropy-21-00549]) is immediate from (45). The “≤” in (51b) is obtained from the same argument of the inequality (34b) and (Equation 35) in the proof of Theorem 2. On the other hand, the equality in (51b) is immediate from $f\left(x\right)g\left(x\right)\geq 0\;\left(\forall x\in {\bar{S}}_{q,\mathrm{opt}}\right)$. The first “=” in (51c) follows from the definition of ${∥\cdot ∥}_{1,{\bar{S}}_{q,\mathrm{opt}}}$, and the “≤” in (51c) follows from the Hölder’s inequality (Equation 9). The final “≤” in (51d) follows from the definition of f(x) in ([Disp-formula FD45a-entropy-21-00549]), in other words, ${∥f∥}_{\infty ,{\bar{S}}_{q,\mathrm{opt}}}\leq 1$, and the resulting ${∥g∥}_{1,{\bar{S}}_{q,\mathrm{opt}}}$ implies the upper bound of $T_{q}\left[p;\lambda_{q}\right]$ if $\lambda_{q,\mathrm{opt}}$ exists for given *q* and σ.Next, we construct a maximizer $p_{\mathrm{opt}}\left(x\right)$ achieving this bound and show its uniqueness, which is done by checking the conditions where all three “≤” in (51) become “=”. As for the first “≤” in (51b), it becomes “=” if and only if p(x) becomes positive only in ${\bar{S}}_{q,\mathrm{opt}}$. Namely,
(52)$$\begin{array}{c} f\left(x\right)=p^{0}\left(x\right)=0\;\mathrm{or}\;1\;\left(x\in {\bar{S}}_{q,\mathrm{opt}}\right)\;\mathrm{and}\;f\left(x\right)=0\;\;\left(a.e.\;x\in R\backslash {\bar{S}}_{q,\mathrm{opt}}\right), \end{array}$$
in other words, the “≤” in (51b) becomes “<” if the above condition (Equation 52) is violated, which is easily verified from the graph of g(x) and the above argument for the “≤” in (51b). On the other hand, in (51c), the second “≤” becomes “=” if and only if $f\left(x\right)=\mathrm{sgn}\left[g\left(x\right)\right]\;\left(a.e.\;x\in {\bar{S}}_{q,\mathrm{opt}}\right)$ due to Lemma 1 (simply by replacing *S* with ${\bar{S}}_{q,\mathrm{opt}}$, in Lemma 1). The final “≤” in (51d) becomes “=” if and only if ${∥f∥}_{\infty ,{\bar{S}}_{q,\mathrm{opt}}}=1$ in (51c). From these three conditions, *f* is uniquely determined as $f_{*}$ in (Equation 47), and from ([Disp-formula FD45a-entropy-21-00549]) the associated maximizer $p_{\mathrm{opt}}$ for q=0 is obtained as
(53)$$\begin{array}{c} p_{\mathrm{opt}}\left(x\right)\;=\left\{\begin{array}{c} \mathrm{arbitrary}\;\mathrm{positive}\;\mathrm{value}\;\;(x\in {\bar{S}}_{q,\mathrm{opt}})\\ 0\;\;(x\in R\backslash {\bar{S}}_{q,\mathrm{opt}}), \end{array}\right. \end{array}$$
where $p_{\mathrm{opt}}\left(x\right)$ should satisfy ${⟪p_{\mathrm{opt}}\left(x\right)⟫}_{{\bar{S}}_{q,\mathrm{opt}}}=1$. Finally, substituting (Equation 53) into the constraint (3c), we have $⟪\left(x^{2}-\sigma^{2}\right)p_{\mathrm{opt}}^{0}\left(x\right)⟫\;=\;{⟪x^{2}-\sigma^{2}⟫}_{{\bar{S}}_{q,\mathrm{opt}}}=0$, and from (Equation 46) $\lambda_{q,\mathrm{opt}}$ and ${\bar{S}}_{q,\mathrm{opt}}$ are uniquely obtained as
(54)$$\begin{array}{c} \lambda_{q,\mathrm{opt}}=-\frac{1}{2}\;\left(<0\right)\;\mathrm{and}\;{\bar{S}}_{q,\mathrm{opt}}=\left[-\sqrt{3}\sigma ,\sqrt{3}\sigma \right], \end{array}$$
respectively. This shows the uniqueness of the representation of $p_{\mathrm{opt}}$ in (Equation 53). (This exceptional case q=0 is also argued in **T1**, where the same result is obtained through a more direct graphical argument, after proving that any candidate $p_{*}\left(x\right)$ for the maximizer $p_{\mathrm{opt}}\left(x\right)$ is defined only on a simply connected interval $S_{*}$ that is symmetric about the origin O. The proof is straightforward but lengthy, and we omit it here.) □

### 4.4. Proof of Theorem 4

**Proof.** Let *p* be arbitrary feasible solutions to **R2** for q>1, and let $p_{\mathrm{opt}}$ be its optimal solution, which is eventually constructed in (Equation 61). Let $\lambda_{q,\mathrm{opt}}$ be a particular value of the additional parameter $\lambda_{q}$ in **R2**, which is associated with $p_{\mathrm{opt}}$ and is eventually constructed in (Equation 65). First, for any *p* and a particular $\lambda_{q,\mathrm{opt}}$, in (Equation 65), we define *f* and *g* as
(55a)$$\begin{array}{ccc} f(x) & = & p^{q}\left(x\right), \end{array}$$
(55b)$$\begin{array}{ccc} g(x) & = & 1+\lambda_{q,\mathrm{opt}}\left(x^{2}-\sigma^{2}\right)p^{1-q}\left(x\right), \end{array}$$
and we define a set ${\underset{\bar{}}{S}}_{q}$ in R:
(56)$$\begin{array}{c} {\underset{\bar{}}{S}}_{q}=\left\{x\;|\;p^{q}\left(x\right)+\lambda_{q,\mathrm{opt}}\left(x^{2}-\sigma^{2}\right)p\left(x\right)\left(=f\left(x\right)g\left(x\right)\right)>0\right\}. \end{array}$$Next, we introduce a subset Q of the feasible solutions *p*,
(57)$$\begin{array}{c} Q=\left\{p\left(x\right)\;|\;\left(5b\right),\;\left(5c\right),\;\mathrm{and}\;p^{q}\left(x\right)+\lambda_{q,\mathrm{opt}}\left(x^{2}-\sigma^{2}\right)p\left(x\right)\;\geq \;0\;\left(\forall x\in R\right)\right\}, \end{array}$$
which is proved to be non-empty in Section A.1.First, we show that the following holds: if p∈Q,
(58a)$$\begin{array}{ccc} R_{q}\left[p;\lambda_{q}\right] & = & R_{q}\left[p;\lambda_{q,\mathrm{opt}}\right]\;=\;{⟪p^{q}\left(x\right)+\lambda_{q,\mathrm{opt}}\left(x^{2}-\sigma^{2}\right)p\left(x\right)⟫}_{{\underset{\bar{}}{S}}_{q}} \end{array}$$
(58b)$$\begin{array}{cc} = & {⟪\left|p^{q}\left(x\right)+\lambda_{q,\mathrm{opt}}\left(x^{2}-\sigma^{2}\right)p\left(x\right)\right|⟫}_{{\underset{\bar{}}{S}}_{q}}={∥fg∥}_{1,{\underset{\bar{}}{S}}_{q}} \end{array}$$
(58c)$$\begin{array}{cc} \geq & {∥f∥}_{\frac{1}{q},{\underset{\bar{}}{S}}_{q}}{∥g∥}_{\frac{1}{1-q},{\underset{\bar{}}{S}}_{q}}\;, \end{array}$$
where $⟪\cdot ⟫=\int_{-\infty }^{\infty }\cdot \;dx$, ${⟪\cdot ⟫}_{{\underset{\bar{}}{S}}_{q}}=\int_{{\underset{\bar{}}{S}}_{q}}\cdot \;dx$, and ${∥\cdot ∥}_{\alpha ,{\underset{\bar{}}{S}}_{q}}={\left(\int_{{\underset{\bar{}}{S}}_{q}}{\left|\cdot \right|}^{\alpha }dx\right)}^{\frac{1}{\alpha }}$. The first “=” in ([Disp-formula FD58a-entropy-21-00549]) follows from the fact that $R_{q}\left[p;\lambda_{q}\right]=⟪p^{q}\left(x\right)⟫+\lambda_{q}⟪\left(x^{2}-\sigma^{2}\right)p\left(x\right)⟫$ in (7c) is independent from the value of $\lambda_{q}$, since any feasible solution p(x) satisfies $⟪\left(x^{2}-\sigma^{2}\right)p\left(x\right)⟫=0$ in (Equation 6), and the second “=” in ([Disp-formula FD58a-entropy-21-00549]) is immediate from the definitions (Equation 56) and (Equation 57). The first “=” in (58b) is also immediate from (Equation 56). The “≥” in (58c) follows from reverse Hölder’s inequality (Equation 11).If $R_{q}\left[p;\lambda_{q}\right]$ achieves the lower bound, and ${∥f∥}_{\frac{1}{q},{\underset{\bar{}}{S}}_{q}}{∥g∥}_{\frac{1}{1-q},{\underset{\bar{}}{S}}_{q}}$ in (58c) saturates at this bound for $p_{\mathrm{opt}}\;\left(\in Q\right)$ and $\lambda_{q,\mathrm{opt}}$, then, from (58), the following has to be satisfied:
$$\begin{array}{c} R_{q}\left[p_{\mathrm{opt}};\lambda_{q,\mathrm{opt}}\right]={∥f∥}_{\frac{1}{q},{\underset{\bar{}}{S}}_{q}}{∥g∥}_{\frac{1}{1-q},{\underset{\bar{}}{S}}_{q}}=\mathrm{the}\;\mathrm{lower}\;\mathrm{bound}. \end{array}$$Therefore, to construct a candidate $p_{\mathrm{opt}}$ achieving this bound, we consider the condition where the “≥” in (58c) becomes “=”. Namely, we rewrite the equality condition (Equation 12) by using (55):
(59a)$$\begin{array}{c} p^{q}\left(x\right)=A{\left[1+\lambda_{q,\mathrm{opt}}\left(x^{2}-\sigma^{2}\right)p^{1-q}\left(x\right)\right]}^{\frac{q}{1-q}}\;>0\;(x\in {\underset{\bar{}}{S}}_{q}),\;i.e., \end{array}$$
(59b)$$\begin{array}{c} p^{1-q}\left(x\right)=C\left[1+\lambda_{q,\mathrm{opt}}\left(x^{2}-\sigma^{2}\right)p^{1-q}\left(x\right)\right]\;>0\;(x\in {\underset{\bar{}}{S}}_{q}), \end{array}$$
where $C=A^{\frac{1-q}{q}}\;>0$. From (59b), it is immediate that
(60)$$\begin{array}{c} p^{1-q}\left(x\right)\left[1+C\lambda_{q,\mathrm{opt}}\left(\sigma^{2}-x^{2}\right)\right]=C\;\left(x\in {\underset{\bar{}}{S}}_{q}\right), \end{array}$$
and, hence,
$$\begin{array}{c} p\left(x\right)={\left(\frac{1}{C}\right)}^{\frac{1}{q-1}}{\left[1+C\lambda_{q,\mathrm{opt}}\left(\sigma^{2}-x^{2}\right)\right]}^{\frac{1}{q-1}}={\left[\lambda_{q,\mathrm{opt}}\left(\sigma^{2}-x^{2}\right)+\frac{1}{C}\right]}^{\frac{1}{q-1}}\;\left(x\in {\underset{\bar{}}{S}}_{q}\right), \end{array}$$
since $\left[1+C\lambda_{q,\mathrm{opt}}\left(\sigma^{2}-x^{2}\right)\right]>0$ because C>0 and $p^{1-q}\left(x\right)>0\;(x\in {\underset{\bar{}}{S}}_{q}$) in (Equation 60). On the other hand, for p∈Q, it is also immediate that $p\left(x\right)=0\;\left(x\in R\backslash {\underset{\bar{}}{S}}_{q}\right)$ from (Equation 56) and (Equation 57). Thereby, a candidate of the minimizer (in Q) is constructed as:
(61)$$\begin{array}{c} p_{\mathrm{opt}}\left(x\right)=\left\{\begin{array}{c} {\left[\lambda_{q,\mathrm{opt}}\left(\sigma^{2}-x^{2}\right)+\frac{1}{C}\right]}^{\frac{1}{q-1}}=\frac{1}{Z_{q}^{*}}\exp_{q^{*}}\left(-\beta_{q}^{*}x^{2}\right)\;\;\left(x\in {\underset{\bar{}}{S}}_{\mathrm{opt}}\right)\\ 0\;\;(x\in R\backslash {\underset{\bar{}}{S}}_{\mathrm{opt}}), \end{array}\right. \end{array}$$
where $q^{*}=2-q$. (The distribution in (Equation 61) is called the $q^{*}$-Gaussian [31].) In (Equation 61), ${\underset{\bar{}}{S}}_{\mathrm{opt}}\;=\;\left[-\sqrt{\sigma^{2}+\frac{1}{C\lambda_{q,\mathrm{opt}}}},\sqrt{\sigma^{2}+\frac{1}{C\lambda_{q,\mathrm{opt}}}}\;\right]$ is now specified and $\lambda_{q,\mathrm{opt}}$, $Z_{q}^{*}$, and $\beta_{q}^{*}$ are uniquely determined as $\lambda_{q,\mathrm{opt}}=\frac{1}{2C\sigma^{2}}\frac{q-1}{q}>0$, $Z_{q}^{*}=\sqrt{\frac{3q-1}{q-1}}\sigma B\left(\frac{q}{q-1},\frac{1}{2}\right)$, and $\beta_{q}^{*}=\frac{1}{\left(3q-1\right)\sigma^{2}}$, respectively, as shown in the following calculations. We note the formula ([42], p. 253):
$$\begin{array}{c} \int_{0}^{1}x^{\alpha }{\left(1-x^{\gamma }\right)}^{\beta }dx=\frac{1}{\gamma }B\left(1+\beta ,\;\frac{1+\alpha }{\gamma }\right) \end{array}$$
is repeatedly used for the integrations below. (More precisely, for (Equation 62) we set α=0(>−1), $\beta =\frac{1}{q-1}\;\left(>-1\right)$, and γ=2(>0).) For (Equation 63), we set α=2(>−1), $\beta =\frac{1}{1-q}\;\left(>-1\right)$, and γ=2(>0). Substituting (Equation 61) into the constraint (5b), we have
(62)$$\begin{array}{c} ⟪p_{\mathrm{opt}}\left(x\right)⟫=r^{\left(\frac{2}{q-1}+1\right)}{\lambda_{q,\mathrm{opt}}}^{\frac{1}{q-1}}B\left(\frac{q}{q-1},\frac{1}{2}\right)=1, \end{array}$$
where *r* is defined by $r^{2}\;=\;\sigma^{2}+\frac{1}{\lambda_{q,\mathrm{opt}}C}.\;$ Note that $r^{2}=\frac{3q-1}{q-1}\sigma^{2}>0$ as shown in (Equation 65). On the other hand, substituting (Equation 61) into the constraint (5c), we have
(63)$$\begin{array}{c} ⟪x^{2}p_{\mathrm{opt}}\left(x\right)⟫=r^{\left(\frac{2}{q-1}+3\right)}{\lambda_{q,\mathrm{opt}}}^{\frac{1}{q-1}}B\left(\frac{q}{q-1},\frac{3}{2}\right)=\sigma^{2}. \end{array}$$Substitution of (Equation 62) and (Equation 63) (multiplied with $\sigma^{2}$ to its both sides) into ([Disp-formula FD7a-entropy-21-00549]) yields
(64)$$\begin{array}{c} r^{2}B\left(\frac{q}{q-1},\frac{3}{2}\right)=\sigma^{2}B\left(\frac{q}{q-1},\frac{1}{2}\right),\;i.e.,\;r^{2}=\frac{3q-1}{q-1}\sigma^{2}\left(=\sigma^{2}+\frac{1}{\lambda_{q,\mathrm{opt}}C}\right), \end{array}$$
in which $B\left(x,y+1\right)=\frac{y}{x+y}B\left(x,y\right)$ ([42], p. 254) is used. Thus, $\lambda_{q,\mathrm{opt}}$ is obtained as
(65)$$\begin{array}{c} \lambda_{q,\mathrm{opt}}=\frac{1}{2C\sigma^{2}}\frac{q-1}{q}. \end{array}$$Since
$$\begin{array}{c} r^{2}=\sigma^{2}+\frac{1}{\lambda_{q,\mathrm{opt}}C}=\frac{3q-1}{q-1}\sigma^{2}. \end{array}$$While, from (Equation 62), or (Equation 63), and (Equation 64) that determines *r* with *q* and σ, $\lambda_{q,\mathrm{opt}}\;\left(>0\right)$ is uniquely obtained for any given q(>1) and σ, and hence, *C* is also uniquely determined from (Equation 65):
$$\begin{array}{c} C=\frac{1}{2\sigma^{2}}\frac{q-1}{q}r^{q+1}B^{q-1}\left(\frac{q}{q-1},\frac{1}{2}\right). \end{array}$$Next, we obtain $Z_{q}^{*}$ and $\beta_{q}^{*}$ from (Equation 61) and (Equation 65):
$$\begin{array}{c} \begin{array}{cc} {\left[\lambda_{q,\mathrm{opt}}\left(\sigma^{2}-x^{2}\right)+\frac{1}{C}\right]}^{\frac{1}{q-1}} & ={\left(\frac{3q-1}{2Cq}\right)}^{\frac{1}{q-1}}{\left[1-\frac{1-q_{*}}{\left(3q-1\right)\sigma^{2}}x^{2}\right]}^{\frac{1}{q_{*}}}\\ & =\frac{1}{Z_{q}^{*}}\exp_{q_{*}}\left[-\frac{x^{2}}{\left(3q-1\right)\sigma^{2}}\right], \end{array} \end{array}$$
which yields
$$\begin{array}{c} \beta_{q}^{*}=\frac{1}{\left(3q-1\right)\sigma^{2}},Z_{q}^{*}={\left(\frac{3q-1}{2Cq}\right)}^{\frac{1}{1-q}}=\sqrt{\frac{3q-1}{q-1}}\sigma B\left(\frac{q}{q-1},\frac{1}{2}\right). \end{array}$$Therefore, from (Equation 61), (Equation 62), and (Equation 65), $p_{\mathrm{opt}}\left(x\right)$ is uniquely determined as in (Equation 61). Note that $p_{\mathrm{opt}}\left(x\right)\in Q$, since
$$\begin{array}{c} 1+\lambda_{q,\mathrm{opt}}\left(x^{2}-\sigma^{2}\right)p_{\mathrm{opt}}^{1-q}\left(x\right)=\frac{1}{C}{\left[\lambda_{q,\mathrm{opt}}\left(\sigma^{2}-x^{2}\right)+\frac{1}{C}\right]}^{-1}>\;0\;\left(x\in {\underset{\bar{}}{S}}_{\mathrm{opt}}\right) \end{array}$$
is immediate from (Equation 61).Next, to prove that the candidate in (Equation 61) is the unique minimizer, we directly compare $⟪p^{q}\left(x\right)⟫$ and $⟪p_{\mathrm{opt}}^{q}\left(x\right)⟫$ in the following way:
$$\begin{array}{c} ⟪p^{q}\left(x\right)⟫-⟪p_{\mathrm{opt}}^{q}\left(x\right)⟫ \end{array}$$
(66a)$$\begin{array}{cc} = & ⟪p^{q}\left(x\right)-p_{\mathrm{opt}}^{q}\left(x\right)+q\lambda_{q,\mathrm{opt}}\left(x^{2}-\sigma^{2}\right)\left[p\left(x\right)-p_{\mathrm{opt}}\left(x\right)\right]⟫-\frac{q}{C}⟪p\left(x\right)-p_{\mathrm{opt}}\left(x\right)⟫ \end{array}$$
(66b)$$\begin{array}{cc} = & {⟪p^{q}\left(x\right)+q\lambda_{q,\mathrm{opt}}\left(x^{2}-\sigma^{2}\right)p\left(x\right)-\frac{q}{C}p\left(x\right)⟫}_{R\backslash {\underset{\bar{}}{S}}_{\mathrm{opt}}} \end{array}$$
(66c)$$\begin{array}{c} +{⟪p^{q}\left(x\right)-p_{\mathrm{opt}}^{q}\left(x\right)-qp_{\mathrm{opt}}^{q-1}\left(x\right)\left[p\left(x\right)-p_{\mathrm{opt}}\left(x\right)\right]⟫}_{{\underset{\bar{}}{S}}_{\mathrm{opt}}}. \end{array}$$Note that ([Disp-formula FD66a-entropy-21-00549]) follows from $⟪p_{\mathrm{opt}}\left(x\right)⟫=⟪p(x)⟫=1$ in (5b) and $⟪x^{2}p_{\mathrm{opt}}\left(x\right)⟫=⟪x^{2}p\left(x\right)⟫=\sigma^{2}$ in (5c), (66b) follows from (Equation 61), in other words, $p_{\mathrm{opt}}\left(x\right)=0\;\left(x\in R\backslash {\underset{\bar{}}{S}}_{\mathrm{opt}}\right)$, and (66c) follows from ([Disp-formula FD66a-entropy-21-00549]) by using $q\lambda_{q,\mathrm{opt}}\left(x^{2}-\sigma^{2}\right)-\frac{q}{C}=-qp_{\mathrm{opt}}^{q-1}\left(x\right)\;\left(x\in {\underset{\bar{}}{S}}_{\mathrm{opt}}\right)$, which is immediate from (Equation 61). Finally, substituting $\lambda_{q,\mathrm{opt}}=\frac{1}{2C\sigma^{2}}\frac{q-1}{q}$ in (Equation 65) into (66b), we obtain
$$\begin{array}{c} {⟪p^{q}\left(x\right)+\frac{q-1}{2C\sigma^{2}}p\left(x\right)\left(x^{2}-\frac{3q-1}{q-1}\sigma^{2}\right)⟫}_{R\backslash {\underset{\bar{}}{S}}_{\mathrm{opt}}}\geq 0, \end{array}$$
in which the equality holds if and only if $p\left(x\right)\;=\;0\;(x\in R\backslash {\underset{\bar{}}{S}}_{\mathrm{opt}})$, since
$$\begin{array}{c} {\underset{\bar{}}{S}}_{\mathrm{opt}}=\left[-\sqrt{\sigma^{2}+\frac{1}{C\lambda_{q,\mathrm{opt}}}},\sqrt{\sigma^{2}+\frac{1}{C\lambda_{q,\mathrm{opt}}}}\;\right]=\left[-\sqrt{\frac{3q-1}{q-1}}\sigma ,\sqrt{\frac{3q-1}{q-1}}\sigma \right], \end{array}$$
and hence,
$$\begin{array}{c} \frac{q-1}{2C\sigma^{2}}\left(x^{2}-\frac{3q-1}{q-1}\sigma^{2}\right)>0\;\left(x\in R\backslash {\underset{\bar{}}{S}}_{\mathrm{opt}}\right). \end{array}$$On the other hand, the term (66c) can be expressed as
(67)$$\begin{array}{c} {⟪p_{\mathrm{opt}}^{q}\left(x\right)\left\{{\left[\frac{p(x)}{p_{\mathrm{opt}}\left(x\right)}\right]}^{q}-q\frac{p(x)}{p_{\mathrm{opt}}\left(x\right)}+q-1\right\}⟫}_{{\underset{\bar{}}{S}}_{\mathrm{opt}}}. \end{array}$$Because, for q>1, $h\left(X\right)\;=\;X^{q}-qX+q-1\geq 0$ for any X≥0, and because h(X)=0 only when X=1, (Equation 67) is nonnegative and it becomes 0 if and only if X=1, in other words, $p\left(x\right)\;=\;p_{\mathrm{opt}}\left(x\right)$ ($x\in {\underset{\bar{}}{S}}_{\mathrm{opt}}$). This proves that $p_{\mathrm{opt}}\left(x\right)$ in (Equation 61) is the unique minimizer to **R2** for q>1. □

### 4.5. Proof of Theorem 5

**Proof.** Let *p* be arbitrary feasible solutions to **R2** for $\frac{1}{3}<q<1$, and let $p_{\mathrm{opt}}$ be its optimal solution, which is eventually constructed in (Equation 72). Let $\lambda_{q,\mathrm{opt}}$ be a particular value of the additional parameter $\lambda_{q}$ in **R2**, which is associated with $p_{\mathrm{opt}}$ and is eventually constructed in (Equation 76). Then, for any *p* and a particular $\lambda_{q,\mathrm{opt}}$, in (Equation 76), we define *f* and *g* as
(68a)$$\begin{array}{ccc} f(x) & = & p^{q}\left(x\right), \end{array}$$
(68b)$$\begin{array}{ccc} g(x) & = & 1+\lambda_{q,\mathrm{opt}}\left(x^{2}-\sigma^{2}\right)p^{1-q}\left(x\right). \end{array}$$First, we show the following holds for any feasible solution *p*,
(69a)$$\begin{array}{ccc} R_{q}\left[p;\lambda_{q}\right] & = & R_{q}\left[p;\lambda_{q,\mathrm{opt}}\right]=⟪fg⟫ \end{array}$$
(69b)$$\begin{array}{cc} \leq & {∥fg∥}_{1}\leq {∥f∥}_{\frac{1}{q}}{∥g∥}_{\frac{1}{1-q}} \end{array}$$
(69c)$$\begin{array}{cc} = & {∥g∥}_{\frac{1}{1-q}}. \end{array}$$The first “=” in ([Disp-formula FD69a-entropy-21-00549]) follows from the fact that $R_{q}\left[p;\lambda_{q}\right]=⟪p^{q}\left(x\right)⟫+\lambda_{q}⟪\left(x^{2}-\sigma^{2}\right)p\left(x\right)⟫$ in (7c) is independent from the value of $\lambda_{q}$, since any feasible solution p(x) satisfies $⟪\left(x^{2}-\sigma^{2}\right)p\left(x\right)⟫=0$ in (Equation 6), and the second “=” in ([Disp-formula FD69a-entropy-21-00549]) is immediate from (68). The first “≤” in (69b) follows from f(x)g(x)≤|f(x)g(x)|(∀x∈R), since f(x) is always nonnegative but g(x) in (68b) can be negative on some intervals in R by choosing certain p(x). The second “≤” in (69b) follows from Hölder’s inequality (Equation 9). The final “=” in (69c) follows from ${∥f∥}_{\frac{1}{q}}={∥p^{q}∥}_{\frac{1}{q}}={⟪\left|p(x)\right|⟫}^{q}=1$ in (69b).Next, if $R_{q}\left[p;\lambda_{q}\right]$ achieves the upper bound, and ${∥g∥}_{\frac{1}{1-q}}$ in (69c) saturates at this bound for $p_{\mathrm{opt}}$ and $\lambda_{q,\mathrm{opt}}$, then from (69) the following has to be satisfied:
$$R_{q}\left[p_{\mathrm{opt}};\lambda_{q,\mathrm{opt}}\right]={∥fg∥}_{1}={∥f∥}_{\frac{1}{q}}{∥g∥}_{\frac{1}{1-q}}=\;\mathrm{the}\mathrm{upper}\mathrm{bound}.$$Therefore, to construct a candidate $p_{\mathrm{opt}}$ achieving this bound, we consider the condition where the two “≤” in (69) become “=”. As for the first “≤” in (69b), it becomes “=” if g(x)>0(a.e.x∈R), in other words,
(70)$$\begin{array}{c} 1+\lambda_{q,\mathrm{opt}}\left(x^{2}-\sigma^{2}\right)p^{1-q}\left(x\right)>0\;\;\;\;\left(a.e.\;x\in R\right), \end{array}$$
which is eventually verified in (Equation 78). On the other hand, the second “≤” in (69b) becomes “=” if and only if the equality condition (Equation 10) is satisfied, $Ap(x)=B|1+\lambda_{q,\mathrm{opt}}\left(x^{2}-\sigma^{2}\right)p^{1-q}\left(x\right){|}^{\frac{1}{1-q}}$, in other words,
(71)$$\begin{array}{c} p^{1-q}\left(x\right)=C\left|1+\lambda_{q,\mathrm{opt}}\left(x^{2}-\sigma^{2}\right)p^{1-q}\left(x\right)\right|\;\left(\forall x\in R\right), \end{array}$$
where C=${\left(B/A\right)}^{1-q}$>0. Note that A≠ 0 and B≠0 because of (Equation 70). From (Equation 70) and (Equation 71), a candidate of the maximizer is uniquely constructed:
(72)$$\begin{array}{c} p_{\mathrm{opt}}\left(x\right)={\left[\lambda_{q,\mathrm{opt}}\left(\sigma^{2}-x^{2}\right)+\frac{1}{C}\right]}^{\frac{-1}{1-q}}=\frac{1}{Z_{q}^{*}}\exp_{q^{*}}\left(-\beta_{q}^{*}x^{2}\right)\;\;\left(\forall x\in R\right), \end{array}$$
in which *C* =${\left(B/A\right)}^{1-q}$, $\lambda_{q,\mathrm{opt}}=\frac{1}{2C\sigma^{2}}\frac{q-1}{q}\;\left(\;<\;0\right)$, $Z_{q}^{*}=\sqrt{\frac{3q-1}{1-q}}\sigma B\left(\frac{1}{1-q}-\frac{1}{2},\frac{1}{2}\right)$, and $\beta_{q}^{*}=\frac{1}{\left(3q-1\right)\sigma^{2}}$ are uniquely determined, and $\lambda_{q,\mathrm{opt}}\left(\sigma^{2}-x^{2}\right)+\frac{1}{C}>0$ is verified, as shown in the following calculations. We note the formula ([42], p. 253):
$$\begin{array}{c} \int_{0}^{\infty }\frac{dx}{x^{\alpha }{\left(1+x^{\gamma }\right)}^{\beta }}=\frac{1}{\gamma }B\left(\beta -\frac{1-\alpha }{\gamma },\;\frac{1-\alpha }{\gamma }\right) \end{array}$$
is repeatedly used for the integrations below. (More precisely, for (Equation 73) we set α=0(<1), $\beta =\frac{1}{1-q}\;\left(>0\right)$, γ=2(>0), and βγ>1−α is satisfied.) For (Equation 74), we set α=−2(<1), $\beta =\frac{1}{1-q}\;\left(>0\right)$, γ=2(>0), and βγ>1−α is satisfied. Substituting (Equation 72) into the constraint (5b), we have
(73)$$\begin{array}{c} ⟪p_{\mathrm{opt}}\left(x\right)⟫=r^{\left(\frac{2}{q-1}+1\right)}{\left(-\lambda_{q,\mathrm{opt}}\right)}^{\frac{1}{q-1}}B\left(\frac{1}{1-q}-\frac{1}{2},\frac{1}{2}\right)=1, \end{array}$$
where *r* is defined by $r^{2}\;=\;-\left(\sigma^{2}+\frac{1}{\lambda_{q,\mathrm{opt}}C}\right).$ Note that $r^{2}=\frac{3q-1}{1-q}\sigma^{2}>0$ as shown in (Equation 76). On the other hand, substituting (Equation 72) into the constraint (5c), we have
(74)$$\begin{array}{c} ⟪x^{2}p_{\mathrm{opt}}\left(x\right)⟫=r^{\left(\frac{2}{q-1}+3\right)}{\left(-\lambda_{q,\mathrm{opt}}\right)}^{\frac{1}{q-1}}B\left(\frac{1}{1-q}-\frac{3}{2},\frac{3}{2}\right)=\sigma^{2}. \end{array}$$Substitution of (Equation 73) and (Equation 74) after multiplying $\sigma^{2}$ to both sides into ([Disp-formula FD7a-entropy-21-00549]) yields
(75)$$\begin{array}{c} r^{2}B\left(\frac{1}{1-q}-\frac{3}{2},\frac{3}{2}\right)=\sigma^{2}B\left(\frac{1}{1-q}-\frac{1}{2},\frac{1}{2}\right),\;i.e.,\;r^{2}=\frac{3q-1}{1-q}\sigma^{2}, \end{array}$$
in which $B\left(x+1,y-1\right)=\frac{x}{y-1}B\left(x,y\right)$ ([42], p. 254) is used. Thus, we first obtain $\lambda_{q,\mathrm{opt}}$ as
(76)$$\begin{array}{c} \lambda_{q,\mathrm{opt}}=\frac{1}{2C\sigma^{2}}\frac{q-1}{q}, \end{array}$$
since
$$\begin{array}{c} r^{2}=-\left(\sigma^{2}+\frac{1}{\lambda_{q,\mathrm{opt}}C}\right)=\frac{3q-1}{1-q}\sigma^{2}. \end{array}$$Meanwhile, from (Equation 73), or (Equation 74), and (Equation 75) that determines *r* with *q* and σ, $\lambda_{q,\mathrm{opt}}$ (<0) is uniquely determined for any *q* (<1) and σ, and hence, *C* (>0) is also uniquely determined from (Equation 76):
$$\begin{array}{c} C=\frac{1}{2\sigma^{2}}\frac{1-q}{q}r^{q+1}B^{q-1}\left(\frac{1}{1-q}-\frac{1}{2},\frac{1}{2}\right). \end{array}$$Next, we obtain $Z_{q}^{*}$ and $\beta_{q}^{*}$ from (Equation 72) and (Equation 76):
$$\begin{array}{ccc} {\left[\lambda_{q,\mathrm{opt}}\left(\sigma^{2}-x^{2}\right)+\frac{1}{C}\right]}^{\frac{-1}{1-q}} & = & {\left(\frac{3q-1}{2Cq}\right)}^{\frac{1}{q-1}}{\left[1-\frac{1-q_{*}}{\left(3q-1\right)\sigma^{2}}x^{2}\right]}^{\frac{1}{1-q_{*}}}\\ & = & \frac{1}{Z_{q}^{*}}\exp_{q^{*}}\left[-\frac{x^{2}}{\left(3q-1\right)\sigma^{2}}\right], \end{array}$$
which yields
$$\begin{array}{c} \beta_{q}^{*}=\frac{1}{\left(3q-1\right)\sigma^{2}},\;Z_{q}^{*}={\left(\frac{3q-1}{2Cq}\right)}^{\frac{1}{1-q}}=\sqrt{\frac{3q-1}{1-q}}\sigma B\left(\frac{1}{1-q}-\frac{1}{2},\frac{1}{2}\right). \end{array}$$Now, we verify (Equation 70) is satisfied by $p_{\mathrm{opt}}$. Note that if $\frac{1}{3}<q<1$, using (Equation 76) we have
(77)$$\begin{array}{c} \lambda_{q,\mathrm{opt}}\left(\sigma^{2}-x^{2}\right)+\frac{1}{C}=\frac{1}{C}\left(\frac{1-q}{2\sigma^{2}q}x^{2}+\frac{3q-1}{2q}\right)>0 \end{array}$$
and hence (Equation 70) is satisfied by $p_{\mathrm{opt}}$:
(78)$$\begin{array}{c} 1+\lambda_{q,\mathrm{opt}}\left(x^{2}-\sigma^{2}\right)p_{\mathrm{opt}}^{1-q}\left(x\right)=\frac{1}{C}{\left[\lambda_{q,\mathrm{opt}}\left(\sigma^{2}-x^{2}\right)+\frac{1}{C}\right]}^{-1}>0\;\;\;\;\left(a.e.\;x\in R\right), \end{array}$$
which is immediate from (Equation 71), (Equation 72), and (Equation 77). Thus, $p_{\mathrm{opt}}$ is uniquely determined as in (Equation 72).Finally, to prove that this candidate (Equation 72) is the unique maximizer, we directly compare $⟪p^{q}\left(x\right)⟫$ and $⟪p_{\mathrm{opt}}^{q}\left(x\right)⟫$ as follows. Similar to (66), the following holds here:
$$\begin{array}{c} ⟪p^{q}\left(x\right)⟫-⟪p_{\mathrm{opt}}^{q}\left(x\right)⟫ \end{array}$$
(79a)$$\begin{array}{cc} = & ⟪p^{q}\left(x\right)-p_{\mathrm{opt}}^{q}\left(x\right)+q\lambda_{q,\mathrm{opt}}\left(x^{2}-\sigma^{2}\right)\left[p\left(x\right)-p_{\mathrm{opt}}\left(x\right)\right]⟫-\frac{q}{C}⟪p\left(x\right)-p_{\mathrm{opt}}\left(x\right)⟫ \end{array}$$
(79b)$$\begin{array}{cc} = & ⟪p^{q}\left(x\right)-p_{\mathrm{opt}}^{q}\left(x\right)-qp_{\mathrm{opt}}^{q-1}\left(x\right)\left[p\left(x\right)-p_{\mathrm{opt}}\left(x\right)\right]⟫. \end{array}$$Note that ([Disp-formula FD79a-entropy-21-00549]) follows from $⟪p_{\mathrm{opt}}\left(x\right)⟫=⟪p(x)⟫=1$ in (5b) and $⟪x^{2}p_{\mathrm{opt}}\left(x\right)⟫=⟪x^{2}p\left(x\right)⟫=\sigma^{2}$ in (5c), and (79b) follows from ([Disp-formula FD79a-entropy-21-00549]) by using $q\lambda_{q,\mathrm{opt}}\left(x^{2}-\sigma^{2}\right)-\frac{q}{C}=-qp_{\mathrm{opt}}^{q-1}\left(x\right)\;\left(x\in R\right)$, which is immediate from (Equation 72). Because for 0<q<1, $h\left(X\right)\;=\;X^{q}-qX+q-1\leq 0$ for any X≥0 and because h(X)=0 only when X=1, (79b) is not positive, and it becomes 0 if and only if X=1, in other words, $p\left(x\right)\;=\;p_{\mathrm{opt}}\left(x\right)$ (x∈R). This proves that $p_{\mathrm{opt}}\left(x\right)$ in (Equation 72) is the unique maximizer to **R2** for $\frac{1}{3}<q<1$. □

## 5. Conclusions and Discussion

We obtained a new insight about a direct link between generalized entropy and Hölder’s inequality, and yet another proof for Rényi–Tsallis entropy maximization; the *q*-Gaussian distribution is directly obtained from the equality condition of Hölder’s inequality, and its optimality is proved by Hölder’s inequality through Moriguti’s argument. The simplicity in the proofs of Tsallis entropy maximization (Theorem 1, 2, and 3) is worth noting; essentially, several lines of inequalities (including Hölder’s inequality) are sufficient for the proof.

As an analogy, what we have described in this study can be explained as mountain climbing; as for Tsallis entropy maximization, the top of the mountain, in other words, the upper/lower bound is clearly seen from the starting point. Namely, the bounds in (24c), (34d), and (51d) are explicitly given by *q* and σ. Therefore, all we need to do is to keep climbing to the top, in other words, to construct a series of inequalities (24), (34), and (51) that saturate at the bound. On the other hand, for Rényi entropy maximization, the top of the mountain is not clearly seen from the starting point. Namely, the upper/lower bound is not given only by *q* and σ but contains p(x), as in (58c) or (69c). Even in such a case, Hölder’s inequality is still useful for finding a peak of the mountain, in other words, it leads to a candidate of the global optimal, and then we verify this candidate is really the top by using a GPS (global positioning system). In addition, this GPS is obtained as in (66) or (79), thanks to Moriguti [5].

Our technique with Hölder’s inequality plus the additional parameter $\lambda_{q}$ can be useful for other inequalities (e.g., Young’s inequality), and it seems an interesting open problem to clarify what sort of optimization problems can be solved from such a technique.

## Figures and Tables

**Figure 1 entropy-21-00549-f001:**
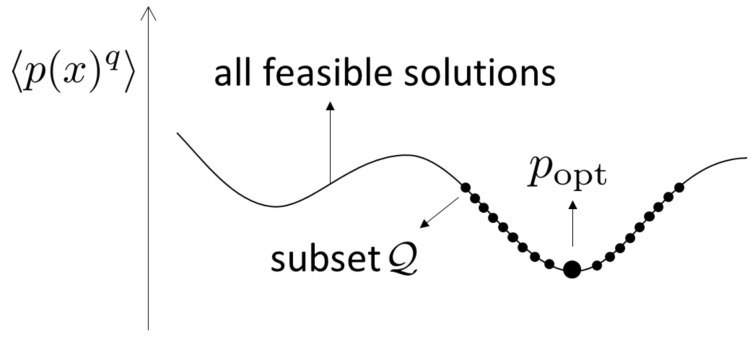
This figure illustrates how our approach for Theorem 4 works in a possible structure of our optimization problem. The whole curve represents all feasible solutions, and the dotted points represent the subset Q in all feasible solutions.

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
