# Peer review of "A Direct Link between Rényi–Tsallis Entropy and Hölder’s Inequality—Yet Another Proof of Rényi–Tsallis Entropy Maximization"

_entropy, 2019, doi:10.3390/e21060549_

Round 1
Reviewer 1 Report
The authors of this paper give alternative constructions of the probability densities maximizing the Tsallis and Renyi differential entropies subject to a second moment constraint (plus a zero moment constraint for the normalization of the density). These proofs are based on equivalent formulations of the problem and the use of the direct and reverse Hölder’s inequalities. Although the results reported are already known, the technique is, to my knowledge, novel and shows the power of an approach based on functional analysis. This method can inspire a similar approach to other optimization problems subject to constraints.
Therefore, the topic is interesting, especially in regard to the tools used. Furthermore, the paper is written in a clear and didactic way. The main results are formulated in Sect. 3 , where the respective proofs are also sketched, while the detailed proofs are shifted to Sect. 4. The proofs are technically demanding and, as I can tell, correct. On these grounds I recommend publication of this paper in Entropy.
To round up their work, I suggest that the authors elaborate a bit on the fact that they consider differential entropies (i.e., continuous probability distributions). It is known from the Shannon case that these entropies have shortcomings such as negative values and non-invariance under coordinate transformations. Actually, Jaynes extended his Maximum Entropy Principle to continuous distributions via the relative entropy instead of the “absolute” entropy.
Misprints.
Line 153. “correspondingle”.
Line 375. The set S_{*} is undefined.
Author Response
We appreciate the referee's kind, constructive comments.
We have realized that we should note only differential entropies are considered in the manuscript, and we have explained this fact in Line 136 - 138 in the revised manuscript.In addition, we thank the referee for pointing out the misprints, which are now corrected in Line 157 and 379.

Reviewer 2 Report
The authors present a new and very elegant proof, how to find the distribution, which maximizes the Tsallis entropy. The result is well known (this is why I gave "Low" assessment for the "Novelty"), though, the simplicity of the proof by using the Hölder inequality makes the paper elegant and original (this is why I rated the presentation as a "High".
After a very thorough introduction, the authors explain their results in form, which is curcumvents most of the mathematical details. This presentation allows the general, interested reader to grasp the main idea, and then dive into the rigorous poof.
I found the paper and its presentation very interesting, and enlightening as the q-Gaussian distribution will is naturally constructed by the proof.
The only, typographic suggestion would be, that the authors should avoid the notation of
<.> for the \int . Because in many fields (e.g. in physics) the <.> is generally used for calculating the expectation value, ie. <.> = \int p\cdot (.) which can lead to confusion. Instead of \langle and \rangle some other symbols like \lmoustache and \rmoustache could be better.
Author Response
First of all, we would like to appreciate the referee's encouraging comments.
As for the typographic suggestion, we have realized that the notation of <.> should be avoided, and we have replaced <.> with <<.>> which is followed by the explanation in Line 133 - 134.

Reviewer 3 Report
The authors develop a new technique for deriving entropy maximising probability distribution functions (PDFs) in the context of different notions of entropy. The entropies of interest in this paper are mainly Tsallis and Renyi entropy, both of which generalise Shannon entropy to a one-parameter family of information measures. The maximisation problem consists of finding a PDF p(x) which maximises Tsallis or Renyi entropy subject to constraints on the PDF such as normalization <p(x)> = 1 and fixed second moment <x^2 p(x)> = const.
The entropy maximising PDFs are already known. However, the paper develops a new tool in order to establish these results in a unifying way. The new tool is Hoelder's inequality, i.e., the statement that the product of two functions in 1-norm is bounded by the product of their \alpha- and \beta-norms (with \alpha^{-1}+\beta^{-1}=1). Essentially, the various entropy maximisation problems considered here can all be reduced to constructing a function that saturates Hoelder's inequality.
The paper is very technical, but well written and clearly structured. The introduction is sufficiently basic to provide a useful overview of the subject. The detailed proofs are only provided after defining all relevant physical and mathematical ingredients and clearly stating the results. The results seem to have been obtained with great care and I trust their correctness. The novel tools that are developed in the paper streamline various previous developments and promise to be useful for future applications. I don't have any substantial suggestions for improvement and recommend the paper for publication.
Author Response
We appreciate the referee's deep understanding. The revised manuscript is attached.
